# SASFT: Sparse Autoencoder-guided Supervised Finetuning to Mitigate Unexpected Code-Switching in LLMs

**Boyi Deng**[1][*]  **Yu Wan**[1][†]  **Baosong Yang**[1]  **Fei Huang**[1]  **Wenjie Wang**[2]  **Fuli Feng**[3][†]

[1]Tongyi Lab, Alibaba Group Inc, [2]National University of Singapore, [3]Anhui Provincial Hospital
dengboyi@mail.ustc.edu.cn    wanyu.wy@alibaba-inc.com
fulifeng93@gmail.com

## Abstract

Large Language Models (LLMs) have impressive multilingual capabilities, but they suffer from unexpected code-switching, also known as language mixing, which involves switching to unexpected languages in the model response. This problem leads to poor readability and degrades the usability of model responses. However, existing work on this issue lacks a mechanistic analysis and shows limited effectiveness. In this paper, we first provide an in-depth analysis of unexpected code-switching using sparse autoencoders and find that when LLMs switch to a language, the features of that language exhibit excessive pre-activation values. Based on our findings, we propose **S**parse **A**utoencoder-guided **S**upervised **F**ine**t**uning (SASFT), which teaches LLMs to maintain appropriate pre-activation values of specific language features during training. Experiments on five models across three languages demonstrate that SASFT consistently reduces unexpected code-switching by more than 50% compared to standard supervised fine-tuning, with complete elimination in one case. Moreover, SASFT maintains or even improves the models' performance on six multilingual benchmarks, showing its effectiveness in addressing code-switching while preserving multilingual capabilities. The code and data are available at `https://github.com/Aatrox103/SASFT`.

## 1 Introduction

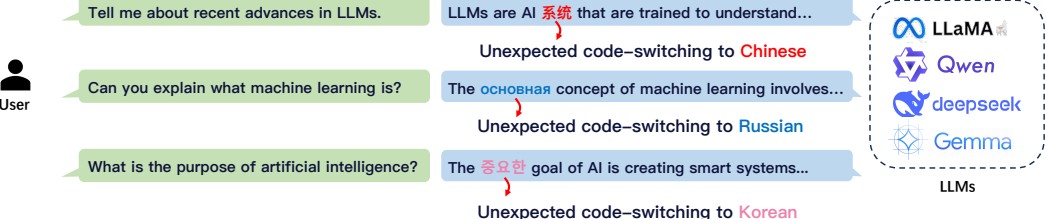

Figure 1: Examples of unexpected code-switching to Chinese, Russian, and Korean.

As the demand for multilingual Large Language Models (LLMs) continues to grow (Qin et al., 2024; Huang et al., 2024; Zhao et al., 2026), researchers seek to improve the multilingual capabilities of LLMs (Team et al., 2024; Grattafiori et al., 2024; Yang et al., 2024). For example, Qwen-3 (Yang et al., 2025) can support 119 languages and performs well on multilingual benchmarks (He et al., 2024a; Zhang et al., 2024; Romanou et al., 2024). In addition, Llama-4 is pre-trained on 200 languages, where over 100 languages have more than 1 billion tokens each (Meta, 2025). Moreover, Gemma-3 offers out-of-the-box support for over 35 languages and pretrained support for over 140 languages (Team et al., 2025). While multilingual capabilities are important for LLMs, they can lead to unexpected code-switching or language mixing (Guo et al., 2025), where LLMs switch to

---

[*]Work done during internships at Alibaba Group.
[†]Corresponding authors.

unexpected languages in their response, as shown in Figure 1. This unexpected code-switching makes it difficult for users to understand and reduces the model's utility (more details please refer to Appendix A). Therefore, addressing unexpected code-switching in LLMs is essential.

To the best of our knowledge, the only attempt to address unexpected code-switching in LLMs is proposed by Guo et al. (2025), who find that DeepSeek-R1 (Guo et al., 2025) suffers from unexpected code-switching and attempt to address it by applying GRPO (Shao et al., 2024) with a language consistency reward. However, their method lacks a deep understanding of unexpected code-switching mechanisms and shows limited effectiveness. This suggests the need for better analysis and solutions.

Inspired by (Deng et al., 2025), which shows that LLMs have language-specific features through sparse autoencoders (SAEs), we conduct preliminary experiments using SAEs and find that unexpected code-switching to a specific language occurs with unusually high pre-activation value of that language's features. Further experiments show that reducing pre-activation values of these language-specific features during inference can mitigate unexpected code-switching. However, this approach requires external intervention and doesn't change the model, without solving the problem fundamentally.

Based on our findings, we propose **S**parse **A**utoencoder-guided **S**upervised **F**ine**t**uning (SASFT) to address unexpected code-switching. The key idea is to teach LLMs to maintain appropriate pre-activation values of irrelevant language features during training, rather than modifying them during inference. Specifically, we introduce an auxiliary loss during supervised fine-tuning (SFT) that encourages the model to keep pre-activation values of specific language features below certain thresholds when generating content in other languages. Since these language features demonstrate strong monolingual characteristics, we aim to reduce code-switching while preserving the model's original capabilities.

Extensive experiments on five widely used models, including the Gemma-2 series (Team et al., 2024), Llama-3.1 series (Meta, 2024), and Qwen-3 series (Yang et al., 2025), demonstrate the effectiveness of our approach. SASFT reduces unexpected code-switching by more than 50% in most cases, with complete elimination (100% reduction) achieved in several scenarios, particularly for the Korean language. Our method significantly outperforms existing methods like GRPO. Notably, SASFT maintains or even improves the models' performance on six multilingual benchmarks, including MMLU (Hendrycks et al., 2021), HumanEval (Peng et al., 2024; Chen et al., 2021), Flores-200 (Goyal et al., 2022; Team et al., 2022), among others. Further analysis reveals that applying SASFT across multiple layers achieves better and more stable results compared to a single layer.

In summary, our main contributions are:

- We provide the first in-depth analysis of unexpected code-switching in LLMs using SAEs, revealing that unexpected code-switching is closely related to unusually high pre-activation of irrelevant language features.

- We propose **S**parse **A**utoencoder-guided **S**upervised **F**ine**t**uning (SASFT), a novel method that addresses unexpected code-switching by teaching LLMs to maintain appropriate pre-activation values of irrelevant language features during training.

- We conduct experiments across five models and six datasets, demonstrating that SASFT effectively reduces unexpected code-switching while maintaining multilingual capabilities.

## 2 PRELIMINARY

**Code-switching reduction.** Code-switching refers to the linguistic phenomenon of alternating between two or more languages within a single text (Poplack, 1978; Kuwanto et al., 2024; Winata et al., 2023). Recent studies of code-switching in LLMs (Zhang et al., 2023; Yong et al., 2023; Huzaifah et al., 2024; Winata et al., 2024; Wang et al., 2025b; Yoo et al., 2024; Li et al., 2024) overlook an important issue: unexpected code-switched content generated by LLMs can confuse users and hinder their comprehension. Therefore, we propose a new task - *Code-Switching Reduction* in LLMs, which aims to minimize unexpected code-switching while preserving the multilingual capabilities of LLMs. Given a multilingual LLM $L$, an unexpected code-switching language $l$, and a set of prompts $\mathcal{X} = \{x_1, x_2, \ldots x_N\}$ where responses should not contain language $l$, the goal of

*Code-Switching Reduction* can be denoted as:

$$\min_{L^*} \frac{1}{N} \sum_{i=1}^{N} \mathbb{I}(CSW(l, P_{L^*}(x_i)))\ s.t.\ Dist(L, L^*) < \epsilon. \tag{1}$$

Here, the function $CSW(l, y)$ checks if text $y$ contains any content in language $l$. $P_{L^*}(x_i)$ is the output when prompting $x_i$ to LLM $L^*$, and $\mathbb{I}(\cdot)$ denotes indicator function. The function $Dist(L, L^*)$ measures the difference between the new LLM $L^*$ and the original LLM $L$. We want to keep this difference small to make sure $L^*$ stays similar to $L$. Since we want to minimize unexpected code-switching while preserving the multilingual capabilities, we use the performance difference on multilingual benchmarks as "distance".

**Code-switching ratio.** We define code-switching ratio as an evaluation metric to measure unexpected language switching in LLM $L$. The ratio can be calculated as:

$$r = \frac{1}{N} \sum_{i=1}^{N} \mathbb{I}(CSW(l, P_L(x_i))). \tag{2}$$

Existing tools cannot reliably detect fine-grained code-switching, such as single characters in another language (Burchell et al., 2024). Thus, we use a script-based approach (see Appendix E.3).

**SAEs.** Sparse Autoencoders (SAEs) are a special type of autoencoder (Hinton & Zemel, 1993). They are used to break down the hidden states of LLMs into a sparse linear combination of learned feature directions. Given a residual stream $\mathbf{x} \in \mathbb{R}^N$ in a certain layer, the SAE calculates a feature activation $\mathbf{a} \in \mathbb{R}^M$, where $M \gg N$. It then uses $\mathbf{a}$ to reconstruct the input as $\hat{\mathbf{x}}$. The typical reconstruction process is described by the following equations:

$$\mathbf{f}(\mathbf{x}) := \mathbf{W}_{\text{enc}}\mathbf{x} + \mathbf{b}_{\text{enc}}, \tag{3}$$
$$\mathbf{a}(\mathbf{x}) := \text{ReLU}(\mathbf{f}(\mathbf{x})), \tag{4}$$
$$\hat{\mathbf{x}}(\mathbf{a}) := \mathbf{W}_{\text{dec}}\mathbf{a} + \mathbf{b}_{\text{dec}}. \tag{5}$$

We focus on the pre-activation value $\mathbf{f}(\mathbf{x})$ rather than the feature activation $\mathbf{a}(\mathbf{x})$, since $\mathbf{a}(\mathbf{x})$ only considers positive values and ignores negative pre-activation values that have meaningful negative projections along feature directions (Mayne et al., 2024). Following the notation of (Rajamanoharan et al., 2024), we define the columns of $\mathbf{W}_{\text{dec}}$ as $\mathbf{d}_i$ for $i = 1, \ldots, M$ and refer to these columns as "features", which can be regarded as specific directions within the residual stream $\mathbf{x}$.

## 3 FEASIBILITY STUDY

### 3.1 UNEXPECTED CODE-SWITCHING IN LLMS

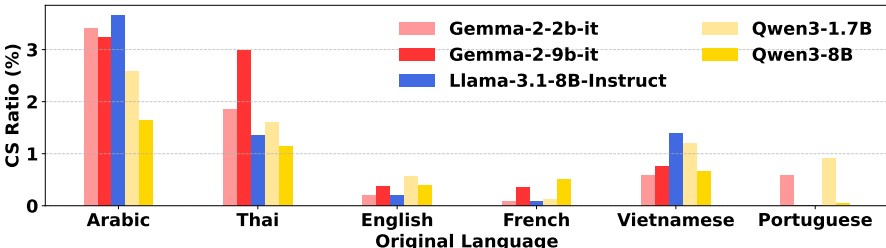

Figure 2: The unexpected code-switching to Chinese for five LLMs in six languages. The results suggest that unexpected code-switching is a common issue in multilingual LLMs.

We intend to investigate whether there are unexpected code-switches to Chinese. To this end, we select queries whose ideal responses should be in a single language without Chinese from six multilingual benchmarks, [1] and generate responses from Gemma-2 (Team et al., 2024), Llama-3.1 (Meta, 2024), and Qwen-3 (Yang et al., 2025). We then measure the unexpected code-switching ratio for Chinese according to Eq. (2). The results are shown in Figure 2, and we observe that: (1) Unexpected code-switching occurs in various LLMs. (2) The ratio of Thai and Arabic content switching to Chinese is higher than others. These findings suggest that unexpected code-switching is a common issue in multilingual LLMs across different languages, and it needs to be addressed.

---

[1]More details in Appendix D.

## 3.2 LANGUAGE-SPECIFIC SAE FEATURES

Deng et al. (2025) revealed that LLMs possess language-specific features—directions in the residual stream that have large projection values only when processing tokens from one particular language. Ablation studies show that removing these features notably impairs the model's performance in the corresponding language while having minimal impact on other languages. Motivated by this, we aim to use these language-specific features to analyze the mechanism behind unexpected code-switching.

## 3.3 UNEXPECTED CODE-SWITCHING IS RELATED TO LANGUAGE-SPECIFIC SAE FEATURES

We aim to explore what causes unexpected code-switching. Inspired by (Deng et al., 2025), we propose that *unexpected code-switching to the target language might be due to unexpectedly high pre-activation values of the target language feature*.

### 3.3.1 PRE-ACTIVATION PATTERN BEFORE CODE-SWITCHING

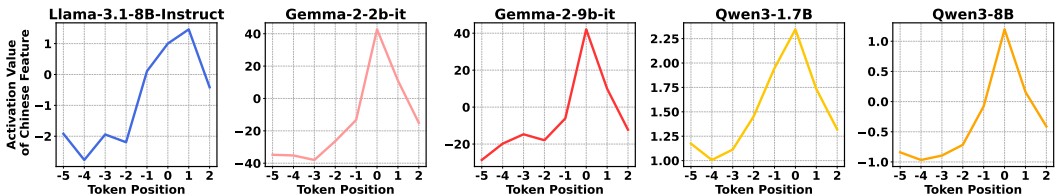

Figure 3: The average pre-activation values of the Chinese feature at different token positions across various LLMs. Position 0 represents the first token that switches to Chinese. Before code-switching occurs, the pre-activation values of the Chinese feature gradually increase.

We collect all the unexpected code-switching responses in Figure 2 and calculate the average pre-activation values of the Chinese feature for different positions near the first token that switches to Chinese, as shown in Figure 3. We observe that the token immediately preceding the first unexpected code-switching token shows higher pre-activation values of the Chinese feature compared to earlier tokens. This indicates that abnormally high pre-activation of features of another language may indicate an upcoming code-switch to that language.

### 3.3.2 ABLATING IRRELEVANT LANGUAGE FEATURE MITIGATES CODE SWITCHING

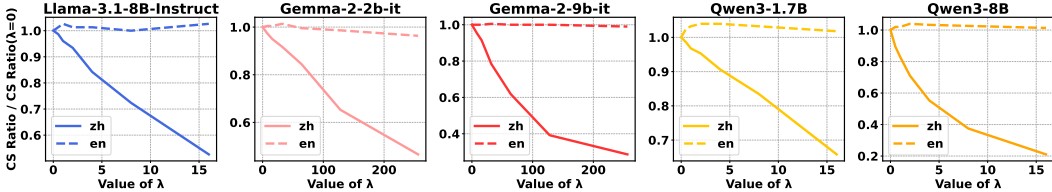

Figure 4: The code-switching ratio to Chinese after ablating Chinese or English features with different $\lambda$. (1) Ablating the Chinese feature can reduce the unexpected code-switching ratio. (2) A higher coefficient $\lambda$ leads to better reduction in the unexpected code-switching ratio. (3) Ablating the English feature has little impact on the unexpected code-switching ratio to Chinese.

In Section 3.3.1, we show that unexpected code-switching might be related to high pre-activation values of language features. Here, we investigate how language features impact unexpected code-switching. Specifically, we use *directional ablation* (Ferrando et al., 2024; Arditi et al., 2024) to subtract the language feature from the residual stream $\mathbf{x} \in \mathbb{R}^N$ at the final layer of the token immediately preceding the first unexpected code-switching token. This process can be expressed as:

$$\mathbf{x}' \leftarrow \mathbf{x} - \lambda \mathbf{d}, \tag{6}$$

where $\mathbf{d}$ represents the language feature and $\lambda$ is the coefficient that controls the degree of ablation. After obtaining $\mathbf{x}'$, we replace $\mathbf{x}$ with $\mathbf{x}'$ and continue the forward pass of the LLMs. We report the

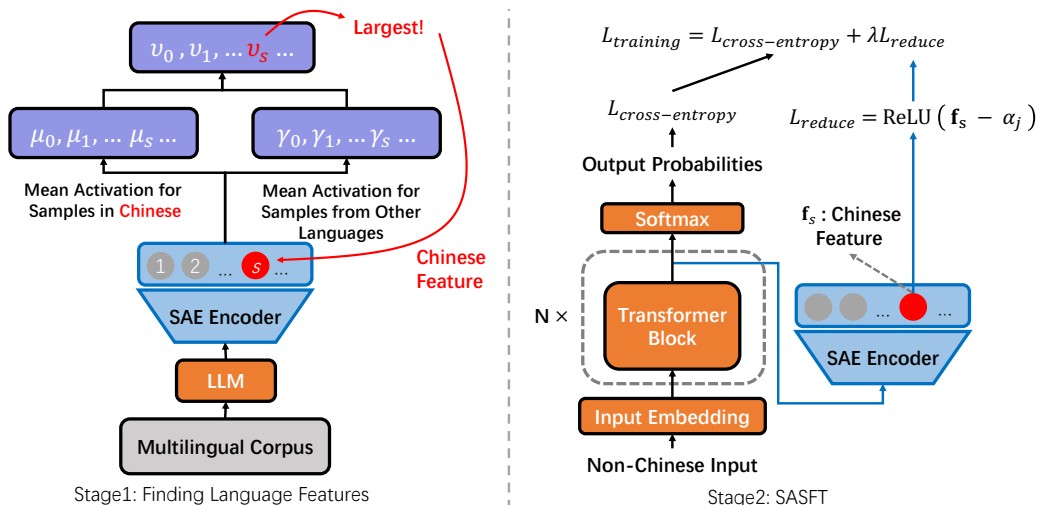

Figure 5: SASFT operates in two steps: First, it identifies language-specific features in LLMs (left), then leverages these features as training signals to reduce code-switching behavior (right).

code-switching ratio with different $\lambda$ in Figure 4. Our observations are as follows: (1) Ablating the Chinese feature can reduce the unexpected code-switching ratio. (2) A higher coefficient $\lambda$ leads to better reduction in the unexpected code-switching ratio. (3) Ablating English features has little impact on the unexpected code-switching ratio to Chinese. These results suggest that changing language-specific features can mitigate unexpected code-switching.

# 4 METHOD

SASFT first identifies language-specific features in LLMs, and then uses these features as training signals to reduce code-switching in LLMs, as shown in Figure 5. We first briefly review the process of finding language-specific features used in (Deng et al., 2025) in Section 4.1, and then introduce SASFT for *Code-Switching Reduction* in Section 4.2.

## 4.1 FINDING LANGUAGE-SPECIFIC FEATURES

Deng et al. (2025) propose a metric to measure the monolinguality of a feature. Given sets of residual streams $\mathcal{D} = \{\mathcal{D}_1, \ldots, \mathcal{D}_K\}$ where $\mathcal{D}_i$ contains the residual streams from language $i$ for a certain layer, they compute how differently feature $s$ activates for language $L$ versus other languages. The computation process is as follows:

$$\mu_s^L = \frac{1}{|\mathcal{D}_L|} \sum_{\mathbf{x} \in \mathcal{D}_L} \mathbf{a}_s(\mathbf{x}),$$

$$\gamma_s^L = \frac{1}{|\mathcal{D} \setminus \{\mathcal{D}_L\}|} \sum_{\mathcal{D}_I \in \mathcal{D} \setminus \{\mathcal{D}_L\}} \frac{1}{|\mathcal{D}_I|} \sum_{\mathbf{x} \in \mathcal{D}_I} \mathbf{a}_s(\mathbf{x}),$$

$$\nu_s^L = \mu_s^L - \gamma_s^L, \tag{7}$$

where $\mathbf{a}_s(\mathbf{x})$ is the activation value of feature $s$ for residual stream $\mathbf{x}$. We then calculate $\nu$ for all languages and features. For each language, we sort all features based on their $\nu$ values from highest to lowest. The top-ranked features are identified as "language-specific features."

## 4.2 SASFT

In Section 3.3, we observe that reducing the pre-activation values of language-specific features during inference can help reduce code-switching. However, this approach has drawbacks: (1) To effectively reduce code-switching, we must lower the pre-activation values of specific language features significantly. We believe this is because specific language features aren't just in the final layer; they

appear in earlier layers too. Changing just the final layer does not affect features from previous layers, so a big reduction is needed. But making large changes or modifying multiple layers can harm the model's other abilities (Deng et al., 2025), making this method impractical. (2) This method requires external intervention and doesn't fundamentally change the model, leading to extra overhead and complexity during inference.

Considering the effectiveness of reducing the pre-activation values of specific language features and its drawbacks during inference, we propose a method to teach LLMs when to lower the pre-activation values of these features during the training process. Specifically, we introduce an auxiliary loss during supervised fine-tuning (SFT) to ensure that LLMs keep the pre-activation values of specific language features below a certain threshold across several layers. Formally, consider a language $L$ that we aim to avoid code-switching to. We have sets of residual streams $\mathcal{D} = \{\mathcal{D}_1, \dots, \mathcal{D}_K\}$, where each $\mathcal{D}_i$ contains the residual streams from training data in language $i$ for a specific layer. The auxiliary loss can be defined as follows:

$$L_{\text{reduce}} = \mathbb{E}_{\mathcal{D}_j \sim \mathcal{D} \setminus \{\mathcal{D}_L\}} \left[ \mathbb{E}_{\mathbf{x} \sim \mathcal{D}_j} \left[ \sum_{s \in \mathcal{S}_L} \text{ReLU} \left( \mathbf{f}_s(\mathbf{x}) - \alpha_j \right) \right] \right], \tag{8}$$

where $\mathbf{f}_s(\mathbf{x})$ is the pre-activation values of feature $s$ for the residual stream $\mathbf{x}$. The set $\mathcal{S}_L$ denotes the language-specific features for language $L$. For each feature $s$ in language $j$, we use $\alpha_j$ to represent its pre-estimated average pre-activation value. We don't set $\alpha_j$ to zero because the pre-estimated average pre-activation value can be negative. In such cases, zero would be too large as a baseline value. Additionally, $\mathcal{D}_L$ is the set of residual streams for language $L$, which we exclude because generating language $L$ from language $L$ does not count as code-switching.

For SASFT, we combine two losses to get the final training loss:

$$L_{training} = L_{\text{cross-entropy}} + \lambda L_{\text{reduce}} \tag{9}$$

where $\lambda$ is a hyperparameter we can adjust to control how much $L_{\text{reduce}}$ contributes to the total loss.

Another straightforward idea is to enhance the pre-activation values of original language features, which might reduce the ratio of code-switching from this language to others. However, our experiments in Appendix F show that this method is less effective than reducing the pre-activation values of unexpected language features. Therefore, we mainly focus on the "reducing" approach.

## 5 EXPERIMENTS

### 5.1 EXPERIMENTAL SETTINGS

**Training data.** We study unexpected code-switching to Chinese, Korean, and Russian. Specifically, we construct six SFT datasets using open-source data (see Appendix C for details). For each language (Chinese, Korean, and Russian), we create two datasets: a larger dataset with 210k samples (100k English, 100k target language, 10k others) and a smaller dataset with 110k samples (50k English, 50k target language, 10k others).

**Models.** We use base models for our experiment as they are suitable for further fine-tuning. Our study includes five models of different sizes and series: Gemma-2-2B, Gemma-2-9B (Team et al., 2024), Llama-3.1-8B (Meta, 2024), Qwen3-1.7B-Base, and Qwen3-8B-Base (Yang et al., 2025). For Gemma-2 models, we use SAEs from Gemma Scope (Lieberum et al., 2024), while for Llama-3.1, we use SAEs from Llama Scope (He et al., 2024b). For Qwen3 models, we train our own set of SAEs on the residual stream of each layer.

**Baselines.** We compare our method with three baseline methods. (1) `SFT`: a standard SFT approach trained with the cross-entropy loss. (2) `SFT+GRPO`: following the work of Guo et al. (2025), who use GRPO to handle unexpected code-switching in DeepSeek-R1 (Guo et al., 2025), we apply GRPO (Shao et al., 2024) with a language consistency reward on an SFT-trained model. The language consistency reward is computed as the percentage of target language words in the model's output. We refer to this baseline as SFT+GRPO. (3) `SFT+Penalty`: an SFT variant that augments the cross-entropy objective with an additional penalty term that discourages tokens from a specified non-target language by reducing their predicted probabilities. Details can be found in Appendix B.

**Implementation.** We use identical hyperparameters for SFT and SASFT. For GRPO, we use a total of 10k samples, consisting of 1k samples for each of the 10 languages. Detailed hyperparameter settings can be found in Appendix E.

**Evaluation.** Our evaluation focuses on two key aspects: (1) the code-switching ratio as defined in Eq. 2, and (2) the model's performance on multilingual benchmarks. The code-switching ratio is calculated using the same query set as described in Section 3.1, while the benchmarks include the multilingual versions of MMLU (Hendrycks et al., 2021), HumanEval (Peng et al., 2024; Chen et al., 2021), Flores-200 (Goyal et al., 2022; Team et al., 2022), HellaSwag (Zellers et al., 2019), LogiQA (Liu et al., 2020), IFEval (Zhou et al., 2023), and MGSM (Shi et al., 2022) from pmmeval (Zhang et al., 2024).

## 5.2 MAIN RESULTS

Table 1: Comparison of code-switching ratios (%) across different methods and models. For each target language (Chinese, Russian, and Korean), we train models on two dataset settings: a 210k dataset and a 110k dataset, then evaluate their code-switching ratio to Chinese, Russian, and Korean. **Bold** numbers indicate the best results. Results show SASFT consistently outperforms the baselines, achieving over 50% reduction in most cases.

| Model | Method | Training Data 210k | | | Training Data 110k | | |
|---|---|---|---|---|---|---|---|
| | | CS: any $\to$ zh | CS: any $\to$ ru | CS: any $\to$ ko | CS: any $\to$ zh | CS: any $\to$ ru | CS: any $\to$ ko |
| Gemma-2-2B | SFT (Baseline) | 0.74 | 0.57 | 3.45 | 0.68 | 0.42 | 1.06 |
| | SFT+GRPO | 0.74 (0%) | 0.49 (-14%) | 3.44 (0%) | 0.72 (+5%) | 0.36 (-16%) | 1.08 (+2%) |
| | SFT+Penalty | 0.67 (-10%) | 0.41 (-27%) | 1.18 (-66%) | 0.66 (-4%) | 0.32 (-25%) | 0.70 (-34%) |
| | SASFT | **0.42 (-43%)** | **0.22 (-61%)** | **0.73 (-79%)** | **0.32 (-53%)** | **0.16 (-62%)** | **0.34 (-68%)** |
| Gemma-2-9B | SFT (Baseline) | 0.78 | 0.12 | 0.81 | 0.80 | 0.05 | 0.50 |
| | SFT+GRPO | 0.78 (0%) | 0.16 (+32%) | 0.70 (-14%) | 0.78 (-2%) | 0.04 (-14%) | 0.48 (-5%) |
| | SFT+Penalty | 0.96 (+23%) | 0.11 (-9%) | 0.63 (-22%) | 0.78 (-3%) | 0.07 (+43%) | 0.29 (-42%) |
| | SASFT | **0.41 (-47%)** | **0.01 (-94%)** | **0.13 (-84%)** | **0.34 (-58%)** | **0.01 (-79%)** | **0.24 (-53%)** |
| Llama-3.1-8B | SFT (Baseline) | 1.16 | 0.67 | 0.57 | 0.48 | 0.79 | 0.37 |
| | SFT+GRPO | 0.52 (-55%) | 0.30 (-55%) | 0.37 (-35%) | 0.40 (-16%) | 0.34 (-57%) | 0.95 (+157%) |
| | SFT+Penalty | 0.42 (-64%) | 0.33 (-51%) | 0.54 (-4%) | 0.39 (-20%) | 0.27 (-66%) | 0.31 (-17%) |
| | SASFT | **0.28 (-76%)** | **0.28 (-59%)** | **0.46 (-19%)** | **0.32 (-34%)** | **0.22 (-72%)** | **0.20 (-46%)** |
| Qwen3-1.7B-Base | SFT (Baseline) | 0.81 | 0.19 | 0.36 | 0.68 | 0.19 | 0.23 |
| | SFT+GRPO | 0.66 (-19%) | 0.11 (-42%) | 0.34 (-6%) | 0.68 (0%) | 0.19 (+1%) | 0.20 (-16%) |
| | SFT+Penalty | 0.53 (-35%) | 0.09 (-53%) | 0.06 (-84%) | 0.49 (-28%) | 0.07 (-62%) | 0.06 (-73%) |
| | SASFT | **0.22 (-72%)** | **0.03 (-85%)** | **0.00 (-100%)** | **0.31 (-55%)** | **0.03 (-87%)** | **0.02 (-93%)** |
| Qwen3-8B-Base | SFT (Baseline) | 0.96 | 0.16 | 0.43 | 0.83 | 0.17 | 0.25 |
| | SFT+GRPO | 0.70 (-14%) | 0.09 (-40%) | 0.22 (-27%) | 0.67 (-26%) | 0.06 (-65%) | 0.12 (-20%) |
| | SFT+Penalty | 0.70 (-27%) | 0.12 (-24%) | 0.23 (-47%) | 0.76 (-9%) | 0.08 (-50%) | 0.18 (-27%) |
| | SASFT | **0.66 (-31%)** | **0.07 (-56%)** | **0.07 (-83%)** | **0.62 (-26%)** | **0.07 (-59%)** | **0.05 (-80%)** |

**Code-switching ratio comparison: SASFT consistently reduces code-switching.** We present the results for code-switching ratio to Chinese (zh), Russian (ru), and Korean (ko) in Table 1, and we observe that: (1) SASFT demonstrates superior performance in reducing code-switching across all scenarios, with more than 50% reduction in 23 out of 30 cases compared to the SFT baseline. (2) SASFT consistently outperforms GRPO across different models and languages. While GRPO shows unstable results with both improvements and deteriorations (e.g., +157% for Llama-3.1-8B with Korean), SASFT maintains consistent reductions across all settings. (3) The effectiveness of SASFT is particularly evident in Qwen-3, while also showing significant improvements in other models like Gemma-2, demonstrating its general applicability across model scales. These results demonstrate that SASFT is a robust and effective method for reducing unexpected code-switching in LLMs, consistently outperforming existing approaches while maintaining stability across different languages and model architectures.

**Performance on multilingual benchmarks: SASFT preserves multilingual capabilities.** We evaluate our method on six multilingual benchmarks to assess its impact on the multilingual capabilities of LLMs, as shown in Table 2. The results demonstrate that: (1) SASFT generally maintains or slightly improves model performance across different benchmarks. For instance, Llama-3.1-8B with SASFT shows notable improvements on several tasks, including MMMLU (+3.13), humaneval (+4.14), and hellaswag (+1.07) compared to the SFT baseline. (2) Even for models where slight performance decreases are observed, the degradation is minimal (usually within 1-2%), suggesting that SASFT effectively reduces code-switching while preserving the model's multilingual capabil-

Table 2: Performance comparison on six benchmarks across different methods. We evaluate models trained on the Chinese 110k dataset setting. Results demonstrate that SASFT successfully maintains model capabilities while reducing code-switching, even showing improvements in several cases. The red numbers indicate performance improvements compared to the SFT. More results are provided in Appendix I.

| Model | Method | MMLU | HumanEval | Flores | HellaSwag | LogiQA | IFEval | MGSM |
|---|---|---|---|---|---|---|---|---|
| | | Acc (%) | Acc (%) | Bleu (%) | Acc (%) | Acc (%) | Acc (%) | Acc (%) |
| Gemma-2-2B | SFT | 29.88 | 76.63 | 22.56 | 24.97 | 28.00 | 14.86 | 12.05 |
| | SFT+GRPO | 29.66 (-0.22) | 76.35 (-0.28) | 22.80 (+0.24) | 26.41 (+1.44) | 26.62 (-1.38) | 14.71 (-0.15) | 10.99 (-1.06) |
| | SFT+Penalty | 30.81 (+0.93) | 80.62 (+3.99) | 22.87 (+0.31) | 26.91 (+1.94) | 27.38 (-0.62) | 15.28 (+0.42) | 11.97 (-0.08) |
| | SASFT | 30.24 (+0.36) | 79.09 (+2.46) | 22.28 (-0.28) | 24.75 (-0.22) | 25.75 (-2.25) | 15.18 (+0.32) | 12.24 (+0.19) |
| Gemma-2-9B | SFT | 44.31 | 95.62 | 30.59 | 32.95 | 34.12 | 21.61 | 44.61 |
| | SFT+GRPO | 44.21 (-0.10) | 95.72 (+0.10) | 30.71 (+0.12) | 33.86 (+0.91) | 31.63 (-2.49) | 21.80 (+0.19) | 45.84 (+1.23) |
| | SFT+Penalty | 46.39 (+2.08) | 97.02 (+1.40) | 30.09 (-0.50) | 32.37 (-0.58) | 34.63 (+0.51) | 21.26 (-0.35) | 46.35 (+1.74) |
| | SASFT | 45.91 (+1.60) | 95.67 (+0.05) | 29.41 (-1.18) | 32.18 (-0.77) | 34.38 (+0.26) | 22.44 (+0.83) | 44.96 (+0.35) |
| Llama-3.1-8B | SFT | 29.99 | 87.74 | 22.81 | 32.39 | 32.88 | 20.08 | 19.92 |
| | SFT+GRPO | 29.67 (-0.32) | 85.58 (-2.16) | 22.34 (-0.47) | 28.17 (-4.22) | 32.12 (-0.76) | 18.91 (-1.17) | 22.83 (+2.91) |
| | SFT+Penalty | 29.70 (-0.29) | 85.43 (-2.31) | 24.36 (+1.55) | 28.63 (-3.76) | 30.37 (-2.51) | 20.00 (-0.08) | 15.81 (-4.11) |
| | SASFT | 33.12 (+3.13) | 91.88 (+4.14) | 23.73 (+0.92) | 33.46 (+1.07) | 30.63 (-2.25) | 19.85 (-0.23) | 18.35 (-1.57) |
| Qwen3-1.7B-Base | SFT | 37.47 | 90.29 | 23.70 | 33.53 | 32.38 | 20.27 | 32.91 |
| | SFT+GRPO | 37.80 (+0.33) | 90.48 (+0.19) | 23.45 (-0.25) | 35.74 (+2.21) | 31.37 (-1.01) | 20.19 (-0.08) | 32.67 (-0.24) |
| | SFT+Penalty | 37.78 (+0.31) | 89.13 (-1.16) | 23.55 (-0.15) | 36.24 (+2.71) | 33.00 (+0.62) | 20.44 (+0.17) | 33.60 (+0.69) |
| | SASFT | 38.38 (+0.91) | 89.04 (-1.25) | 23.67 (-0.03) | 33.71 (+0.18) | 32.38 (0.00) | 20.22 (-0.05) | 30.85 (-2.06) |
| Qwen3-8B-Base | SFT | 52.15 | 95.87 | 29.99 | 42.48 | 42.25 | 33.64 | 58.03 |
| | SFT+GRPO | 50.85 (-1.30) | 96.44 (+0.57) | 30.14 (+0.15) | 44.48 (+2.00) | 41.50 (-0.75) | 33.42 (-0.22) | 55.28 (-2.75) |
| | SFT+Penalty | 50.74 (-1.41) | 94.71 (-1.16) | 30.10 (+0.11) | 34.51 (-7.97) | 39.88 (-2.37) | 34.04 (+0.40) | 56.29 (-1.74) |
| | SASFT | 50.09 (-2.06) | 98.27 (+2.40) | 29.97 (-0.02) | 39.60 (-2.88) | 42.75 (+0.50) | 33.91 (+0.27) | 58.45 (+0.42) |

ities. These results indicate that our SASFT method effectively addresses the code-switching issue without substantially affecting the model's overall performance on multilingual tasks; in some cases, SASFT even improves performance.

## 5.3 IN-DEPTH ANALYSIS

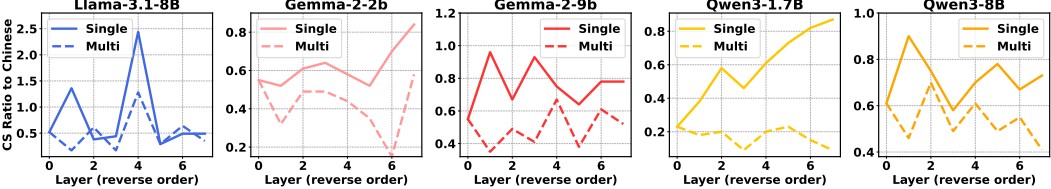

Figure 6: Impact of layer selection on code-switching ratio across different models. Single-layer (solid lines) represents applying SASFT to individual layers, while Multi-layer (dashed lines) represents applying SASFT to consecutive layers starting from the final layer. Layers are counted in reverse order (0 represents the final layer). Results show that multi-layer consistently achieves better and more stable performance than the single-layer approach, while the single-layer effectiveness decreases when moving towards earlier layers.

**Effect of layers used in SASFT: multi-layer outperforms single-layer in reducing code-switching.** We investigate how different layer selections (in reverse order from the final layer) affect SASFT's performance in code-switching reduction, as shown in Figure 6. The results demonstrate that: (1) Multi-layer SASFT consistently shows better performance than the single-layer approach across all models. This is particularly evident in Gemma-2 and Qwen3, where the multi-layer approach (dashed lines) maintains lower code-switching ratios throughout different layer selections. (2) For single-layer SASFT, the performance generally deteriorates as we move towards earlier layers, with the code-switching ratio showing an increasing trend across most models. (3) The impact of layer selection is more pronounced in single-layer interventions, showing higher variability in performance, while multi-layer approaches demonstrate more stable performance across different layer combinations, suggesting better robustness.

**Effect of features used in SASFT: multi-Feature outperforms single-feature in reducing code-switching.** We examine how different feature selection strategies affect SASFT's performance in code-switching reduction, comparing single-feature versus multi-feature approaches across models,

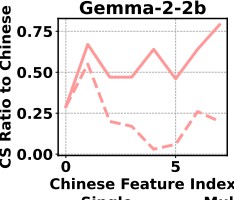 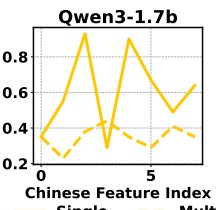 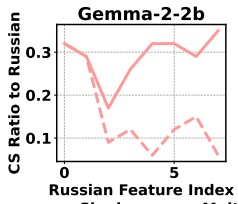 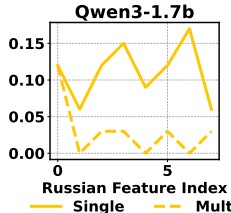

Figure 7: Impact of feature selection on code-switching ratio across different models. Single-feature (solid lines) represents applying SASFT to individual features, while Multi-feature (dashed lines) represents applying SASFT to consecutive features starting from the rank-1 language feature. 0 represents the rank-1 language feature. Results show that multi-feature intervention consistently achieves better and more stable performance than single-feature approach.

as shown in Figure 7. We observe that: (1) Multi-feature SASFT consistently shows better performance than the single-feature approach for Chinese features, maintaining lower code-switching ratios with reduced variance. (2) The performance difference between Chinese and Russian features suggests language-dependent effectiveness, possibly due to models' stronger Chinese language capabilities compared to Russian. (3) Notably, the optimal code-switching reduction is achieved when applying the multi-feature approach.

## 5.4 ABLATION STUDY

To validate the rationality of setting $\alpha_j$ to pre-estimated average values rather than zero in Eq. (8), we compare $\text{SASFT}_{\text{zero}}$ ($\alpha_j = 0$) with SASFT in Table 3. We observe that: (1) $\text{SASFT}_{\text{zero}}$ effectively reduces code-switching and shows comparable performance to SFT+Penalty on Gemma-2-2B, while achieving notably better results on Qwen3-1.7B-Base. (2) SASFT outperforms $\text{SASFT}_{\text{zero}}$ across most configurations, demonstrating that using pre-estimated average pre-activation values is more effective than simply setting them to zero.

Table 3: Comparison of code-switching ratios between different $\alpha_j$ settings. **Bold** numbers indicate the best results while underlined numbers represent the second best. Both $\text{SASFT}_{\text{zero}}$ ($\alpha_j = 0$) and SASFT show effectiveness in reducing code-switching, with SASFT achieving better performance across different settings.

| Model | Method | Training Data 210k | | | Training Data 110k | | |
|---|---|---|---|---|---|---|---|
| | | CS: any $\rightarrow$ zh | CS: any $\rightarrow$ ru | CS: any $\rightarrow$ ko | CS: any $\rightarrow$ zh | CS: any $\rightarrow$ ru | CS: any $\rightarrow$ ko |
| Gemma-2-2B | SFT (Baseline) | 0.74 | 0.57 | 3.45 | 0.68 | 0.42 | 1.06 |
| | SFT+GRPO | 0.74 (0%) | 0.49 (-14%) | 3.44 (0%) | 0.72 (+6%) | 0.32 (-24%) | 1.08 (+2%) |
| | SFT+Penalty | 0.67 (-9%) | 0.41 (-28%) | 1.18 (-66%) | 0.66 (-3%) | 0.28 (-33%) | 0.70 (-34%) |
| | $\text{SASFT}_{\text{zero}}$ | 0.58 (-22%) | 0.90 (+58%) | 1.99 (-42%) | 0.49 (-28%) | 0.36 (-14%) | 0.58 (-45%) |
| | SASFT | **0.42 (-43%)** | **0.22 (-61%)** | **0.73 (-79%)** | **0.32 (-53%)** | **0.16 (-62%)** | **0.34 (-68%)** |
| Qwen3-1.7B-Base | SFT (Baseline) | 0.81 | 0.19 | 0.36 | 0.68 | 0.19 | 0.23 |
| | SFT+GRPO | 0.66 (-19%) | 0.11 (-42%) | 0.34 (-6%) | 0.68 (0%) | 0.19 (0%) | 0.20 (-13%) |
| | SFT+Penalty | 0.53 (-35%) | 0.09 (-53%) | 0.06 (-83%) | 0.49 (-28%) | 0.07 (-63%) | 0.06 (-74%) |
| | $\text{SASFT}_{\text{zero}}$ | 0.49 (-40%) | 0.07 (-63%) | 0.05 (-86%) | 0.40 (-41%) | 0.04 (-79%) | 0.02 (-91%) |
| | SASFT | **0.22 (-73%)** | **0.03 (-84%)** | **0.00 (-100%)** | **0.31 (-54%)** | **0.03 (-84%)** | **0.02 (-91%)** |

## 6 RELATED WORKS

**Code-switching.** Code-switching refers to the linguistic phenomenon of alternating between two or more languages within a single text (Poplack, 1978; Kuwanto et al., 2024; Winata et al., 2023). While recent studies make significant progress in processing code-switching content (Zhang et al., 2023; Yong et al., 2023) and leveraging code-switched data to enhance LLMs (Wang et al., 2025b; Yoo et al., 2024), they overlook a critical issue: unexpected code-switched content generated by LLMs can significantly impair user comprehension. Guo et al. (2025) first attempts to tackle this challenge by applying GRPO (Shao et al., 2024) with a language consistency reward on an SFT-trained model. Recently, Wang et al. (2025a); Nie et al. (2025) show that code-switching closely aligns with that of the model's internal representations.

**SAEs.** SAEs serve as a powerful interpretability tool by decomposing a model's internal representations into meaningful feature directions, enabling researchers to mechanistically explain various phenomena within LLMs (Bricken et al., 2023; Cunningham et al., 2023; Shi et al., 2025). Ferrando et al. (2024) employs SAEs to discover features indicating LLMs' entity recognition, while Cunningham et al. (2023) identifies features associated with apostrophes. Galichin et al. (2025); Fang et al. (2026) use SAEs to identify and validate reasoning features in reasoning models like DeepSeek-R1 (Guo et al., 2025). Particularly noteworthy is the work by Deng et al. (2025), which reveals that certain features are strongly correlated with specific languages, and ablating these features only impacts the model's performance in one language. Inspired by their findings on language-specific features, we employ SAEs to analyze unexpected code-switching behavior and solve it.

## 7 CONCLUSION

We focus on the issue of unexpected code-switching in multilingual LLMs. Through analysis with SAEs, we discover that unexpected code-switching is linked to unusually high pre-activation values of irrelevant language features. Based on this finding, we propose SASFT, a novel approach that guides LLMs to maintain appropriate pre-activation values of language-specific features during training. Extensive experiments on five different models demonstrate that SASFT effectively reduces unexpected code-switching by more than 50% while maintaining or improving performance on various multilingual benchmarks. Our work provides a practical solution for developing more reliable multilingual LLMs, contributing to the advancement of multilingual LLMs.

## ACKNOWLEDGEMENTS

This work was supported by Alibaba Group through Alibaba Research Intern Program.

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

## A  UNEXPECTED CODE-SWITCHING IN LLMS: A GROWING CONCERN

The phenomenon of unexpected code-switching, where language models abruptly switch between different languages during generation, has become increasingly prevalent in various open-source LLMs. This issue significantly impacts user experience and model reliability. For instance, multiple users have reported unexpected code-switching in models like DeepSeek and Qwen, particularly between English and Chinese.

This phenomenon has been widely documented across different community platforms. For DeepSeek, users have reported the code-switching issue both on GitHub, where the model occasionally switches to Chinese mid-conversation[2], and on Reddit, where multiple users experienced random switches to Chinese characters, particularly when generating longer responses[3]. Similar issues have been observed with the Qwen model, where Reddit users reported unexpected Chinese outputs during other language interactions[4].

## B  BASELINE DETAILS

We propose a more intuitive SFT baseline named `SFT+Penalty`. The main idea is to add a penalty term to the SFT loss that minimizes the probabilities of tokens from a certain language.

Specifically, to reduce code-switching to a target language L (e.g., Chinese), we first identify all tokens belonging to language L from the tokenizer, denoted as $V_L$. During training on samples in languages other than L, in addition to the standard cross-entropy loss $L_{CE}$, we add a regularization term at each token position in the response to penalize the model's prediction probability for tokens in $V_L$. The training objective can be formulated as:

$$L_{TLP} = L_{CE} + \beta \cdot \frac{1}{T} \sum_{t=1}^{T} \sum_{v \in V_L} p_\theta(v|x, y_{<t}), \tag{10}$$

where $T$ is the response length, $p_\theta(v|x, y_{<t})$ is the model's probability of predicting token $v$ at response position $t$, and $\beta$ is the penalty coefficient.

---

[2]https://github.com/deepseek-ai/DeepSeek-R1/issues/110.
[3]https://www.reddit.com/r/LocalLLaMA/comments/1i958ii/anyone_else_experienced_deepseek_randomly/.
[4]https://www.reddit.com/r/LocalLLaMA/comments/1hlitkn/qwen_often_output_chinese/.

## C    DETAILS OF SFT TRAINING DATA

We construct six SFT datasets using a variety of open-source data, with the statistics summarized in Table 4 and Table 5. Each dataset represents a distinct setting in which we carefully control the total sample size and language composition. Specifically, in each configuration, the datasets include either approximately 210k or 110k samples, focusing on three target languages: Korean (ko), Russian (ru), and Chinese (zh).

Among the data sources, KULLM[5], Tulu3 (Lambert et al., 2024), WildChat (Zhao et al., 2024), and BelleGroup[6] each provide single-language samples: specifically, KULLM for Korean, Tulu3 for English, WildChat for Russian, and BelleGroup for Chinese. The remaining data, Multialpaca (Wei et al., 2023), Flores (Goyal et al., 2022), and GSM8KInstruct (Cobbe et al., 2021)[7], offer multilingual data, contributing samples across various languages.

Table 4: Number of samples of each language in different dataset settings. Each row shows the distribution of samples across languages for different dataset sizes (either 210k or 110k). For Russian (ru), the sample size is approximate due to limited available data.

| Dataset | Samples per Language | | | | | | | | | | | | | |
|---|---|---|---|---|---|---|---|---|---|---|---|---|---|---|
| | en | ko | vi | zh | th | fr | ar | es | pt | de | ja | id | ru | other |
| ko-210k | 100000 | 100000 | 1000 | 1000 | 1000 | 1000 | 1000 | 1000 | 1000 | 1000 | 1000 | 1000 | 1000 | 2276 |
| ko-110k | 50000 | 50000 | 1000 | 1000 | 1000 | 1000 | 1000 | 1000 | 1000 | 1000 | 1000 | 1000 | 1000 | 2276 |
| ru-210k | 100000 | 1000 | 1000 | 1000 | 1000 | 1000 | 1000 | 1000 | 1000 | 1000 | 1000 | 1000 | 86354 | 2276 |
| ru-110k | 50000 | 1000 | 1000 | 1000 | 1000 | 1000 | 1000 | 1000 | 1000 | 1000 | 1000 | 1000 | 50000 | 2276 |
| zh-210k | 100000 | 1000 | 1000 | 100000 | 1000 | 1000 | 1000 | 1000 | 1000 | 1000 | 1000 | 1000 | 1000 | 2276 |
| zh-110k | 50000 | 1000 | 1000 | 50000 | 1000 | 1000 | 1000 | 1000 | 1000 | 1000 | 1000 | 1000 | 1000 | 2276 |

Table 5: Number of samples from each source in different dataset settings. Each row shows the sample counts contributed by different data sources under various dataset sizes.

| Dataset | Samples per Source | | | | | | |
|---|---|---|---|---|---|---|---|
| | KULLM | Tulu3 | Multialpaca | Flores | GSM8K | BelleGroup | WildChat |
| ko-210k | 97834 | 97697 | 9774 | 4775 | 1250 | 985 | 961 |
| ko-110k | 48892 | 48831 | 8829 | 3691 | 1087 | 985 | 961 |
| ru-210k | 984 | 97697 | 10823 | 4878 | 1628 | 985 | 82635 |
| ru-110k | 984 | 48831 | 9549 | 3768 | 1296 | 985 | 47863 |
| zh-210k | 984 | 97697 | 7854 | 6088 | 1581 | 98111 | 961 |
| zh-110k | 984 | 48831 | 7854 | 4339 | 1266 | 49041 | 961 |

## D    DETAILS OF THE CODE-SWITCHING EVALUATION DATA

For the preliminary experiments presented in Figure 2, we use prompts in Arabic, Thai, English, French, Vietnamese, and Portuguese, totaling 34,996 examples. However, we observe that some of these languages exhibit relatively low code-switching ratios. Consequently, in our subsequent main experiments, we replace these low-ratio languages with alternatives that demonstrate more pronounced code-switching behavior.

For our main experiments, we use prompts from the multilingual versions of MMLU (Hendrycks et al., 2021), MGSM (Shi et al., 2023), HellaSwag (Zellers et al., 2019), LogiQA (Liu et al., 2020), IFEval (Zhou et al., 2023), and Flores-200 (Goyal et al., 2022; Team et al., 2022), all provided by pmmeval (Zhang et al., 2024). In total, our evaluation set comprises 1,756 examples in Chinese (zh), 1,146 in Arabic (ar), and 1,150 examples each in Thai (th), Vietnamese (vi), Korean (ko), and Japanese (ja).

---

[5]https://huggingface.co/datasets/nlpai-lab/kullm-v2.
[6]https://huggingface.co/datasets/BelleGroup/train_0.5M_CN.
[7]https://huggingface.co/datasets/Mathoctopus/GSM8KInstruct_Parallel.

Table 6: Code-switching evaluation dataset: source-to-target language pairs and sample counts.

| CS Target | Prompt Source | # Examples | # Runs | Total Samples |
|---|---|---|---|---|
| zh | ar | 1,146 | 8 | |
| | th | 1,150 | 8 | 27,568 |
| | vi | 1,150 | 8 | |
| ru | ar | 1,146 | 8 | |
| | th | 1,150 | 8 | 27,568 |
| | ko | 1,150 | 8 | |
| ko | zh | 1,756 | 8 | |
| | th | 1,150 | 8 | 32,448 |
| | ja | 1,150 | 8 | |

We investigate code-switching behavior to three target languages: Chinese (zh), Russian (ru), and Korean (ko). For each target language, we evaluate prompts from three different source languages. Table 6 presents the composition of our code-switching evaluation dataset, where each example is tested 8 times to ensure robust detection of code-switching patterns.

# E  IMPLEMENTATION DETAILS

## E.1  TRAINING

We use the Hugging Face TRL library[8] in conjunction with DeepSpeed[9] for SFT, and the combination of TRL and vLLM (Kwon et al., 2023)[10] for GRPO.

For SFT and SASFT, both learning rate and $\lambda$ in Eq. (8) are selected via grid search over respective intervals, with the learning rate ranging from $1 \times 10^{-6}$ to $2 \times 10^{-4}$ and $\lambda$ from $5 \times 10^{-5}$ to $1 \times 10^{-2}$. For SASFT, all main experiment results are obtained using the first two language features from the last two layers. The following table summarizes the optimal hyperparameters and corresponding training times for SFT on 110k samples for each model:

Table 7: Optimal hyperparameters and SFT training time for 110k samples across different models.

| Model | Learning Rate | $\lambda$ | SFT Training Time | Deepspeed Optimization Level |
|---|---|---|---|---|
| Gemma-2-2B | $5.0 \times 10^{-5}$ | $5.0 \times 10^{-4}$ | 1h | None |
| Gemma-2-9B | $5.0 \times 10^{-6}$ | $5.0 \times 10^{-4}$ | 11h | ZeRO2 |
| Llama-3.1-8B | $5.0 \times 10^{-5}$ | $1.0 \times 10^{-3}$ | 3h | ZeRO1 |
| Qwen3-1.7B | $1.0 \times 10^{-4}$ | $1.0 \times 10^{-3}$ | 40min | None |
| Qwen3-8B | $5.0 \times 10^{-5}$ | $5.0 \times 10^{-4}$ | 3.1h | ZeRO1 |

For all experiments, the batch size is set to 256, weight decay to 0.1, warmup steps to 100, and the cosine learning rate scheduler is employed. AdamW (fused) serves as the optimizer, and training is performed using `bf16` precision. Further, for SASFT, we select the last two layers and the first two features. All reported training times correspond to nodes equipped with 8 NVIDIA A100 or H20 GPUs; times may vary based on model size and hardware.

For GRPO, we employ the TRL library in combination with vLLM, conducting a grid search for the learning rate within the range $1 \times 10^{-8}$ to $1 \times 10^{-6}$. We use a batch size of 256 and set the number of rollouts to 8. The following table presents the optimal GRPO learning rates and corresponding training times:

---

[8]https://github.com/huggingface/trl.
[9]https://github.com/deepspeedai/DeepSpeed.
[10]https://github.com/vllm-project/vllm.

Table 8: Optimal GRPO learning rates and training times.

| Model | GRPO Learning Rate | GRPO Training Time |
|---|---|---|
| Gemma-2-2B | $7.0 \times 10^{-7}$ | 40 min |
| Gemma-2-9B | $7.0 \times 10^{-8}$ | 3.5 h |
| Llama-3.1-8B | $5.0 \times 10^{-7}$ | 2 h |
| Qwen3-1.7B | $1.0 \times 10^{-7}$ | 35 min |
| Qwen3-8B | $5.0 \times 10^{-7}$ | 2 h |

All GRPO experiments are performed under similar hardware configurations as SFT, utilizing 8 NVIDIA A100 or H20 GPUs, with training duration depending on model size and hardware specifications.

### E.2 INFERENCE

During inference, we use the following decoding parameters:

- top-p sampling: $0.8$
- repetition penalty: $1.0$
- temperature: $1.0$

To reduce the inference time, we utilize the no-thinking mode for Qwen-3.

### E.3 CODE-SWITCHING DETECTION

We use GlotScript (Kargaran et al., 2024) for code-switching detection. GlotScript identifies different writing systems based on Unicode character ranges. We focus on Chinese, Russian, and Korean because their writing systems (Han, Cyrillic, and Hangul, respectively) are distinct from other scripts. This makes them easily distinguishable, unlike languages such as English and French that share the Latin alphabet and cannot be reliably separated based on script alone.

In our detection process, if Han characters appear in a response that should not contain Chinese, we mark it as unexpected code-switching to Chinese. The same rule applies to Cyrillic and Hangul characters for detecting unexpected code-switching to Russian and Korean, respectively.

## F SASFT VARIANT

### F.1 METHOD

Another idea is that enhancing the pre-activation values of original language features should be able to reduce the ratio of code-switching from this language to other languages. Therefore, we extend Eq. (8) to enhance the pre-activation values of original language features, which can be defined as follows:

$$L_{\text{enhance}} = \mathbb{E}_{\mathbf{x} \sim \mathcal{D}_M} \left[ \sum_{s \in \mathcal{S}_M} \text{ReLU} \left( \beta_M - \mathbf{f}_s(\mathbf{x}) \right) \right], \tag{11}$$

where $M$ is the language intended for enhancement, and $\beta_M$ is the pre-estimated average pre-activation values of feature $s$ in language $M$. We call this variant as $\text{SASFT}_{Enhance}$.

### F.2 EXPERIMENTS

In this section, we focus on $\text{SASFT}_{Enhance}$ which enhance original language features using Eq. 11.

**Code-Switching Ratio Comparison: Our Methods Effectively Reduce Code-Switching.** Table 9 presents code-switching ratios from Arabic and Thai to Chinese, Russian, and Korean. We

observe that SASFT$_{Enhance}$ generally reduces code-switching compared to the SFT baseline, outperforming GRPO in most cases (7 out of 12). Importantly, SASFT$_{Reduce}$ achieves the lowest ratios in all settings, consistently providing the best results. Overall, both enhancement and reduction approaches are effective, with the reduction method showing superior performance.

Table 9: Evaluation of code-switching reduction for Arabic and Thai as enhanced source languages. Models are tested on their tendency to switch from these source languages to Chinese, Russian, and Korean. **Bold** numbers indicate the best results while underlined numbers represent the second best in each column.

| Model | Method | Enhanced Language: ar | | | Enhanced Language: th | | |
|---|---|---|---|---|---|---|---|
| | | CS: ar → zh | CS: ar → ru | CS: ar → ko | CS: th → zh | CS: th → ru | CS: th → ko |
| Gemma-2-2B | SFT (Baseline) | 1.14 | 1.22 | 0.17 | 0.43 | 0.43 | 0.00 (0%) |
| | SFT+GRPO | 0.79 (-31%) | 0.61 (-50%) | 0.09 (-47%) | 0.95 (+121%) | 0.17 (-60%) | 0.00 (0%) |
| | SASFT$_{Enhance}$ | 1.31 (+15%) | 0.70 (-43%) | **0.00 (-100%)** | 0.43 (0%) | 0.26 (-40%) | 0.00 (0%) |
| | SASFT$_{Reduce}$ | **0.61 (-46%)** | **0.26 (-79%)** | **0.00 (-100%)** | 0.17 (-60%) | **0.09 (-79%)** | **0.00 (0%)** |
| Qwen3-1.7B-Base | SFT (Baseline) | 1.04 | 0.26 | 0.26 | 0.35 | 0.18 | 0.09 |
| | SFT+GRPO | 0.61 (-41%) | 0.26 (0%) | 0.26 (0%) | 0.53 (+51%) | 0.09 (-50%) | 0.09 (0%) |
| | SASFT$_{Enhance}$ | 0.26 (-75%) | 0.17 (-35%) | 0.09 (-65%) | 0.44 (+26%) | 0.17 (-6%) | 0.09 (0%) |
| | SASFT$_{Reduce}$ | **0.17 (-84%)** | **0.00 (-100%)** | **0.00 (-100%)** | **0.26 (-26%)** | **0.00 (-100%)** | **0.00 (-100%)** |

## G LIMITATIONS AND FUTURE WORK

Our study has several limitations that we plan to address in future work: First, we only explore unexpected code-switching to Chinese, Russian, and Korean. Adding more languages would make the study more complete. Second, while we experiment with 5 LLMs from 3 model families of different sizes, all models are under 9B. Testing on larger models would provide a more comprehensive understanding of our method's effectiveness. Third, theoretically, our method only requires constraints on the model's hidden states, so it should be possible to extend it to other fine-tuning approaches like DPO and GRPO. We believe this is a promising direction for future research. Finally, although using pre-estimated average pre-activation values as thresholds works well in our experiments, finding a fine-grained token-level threshold could potentially improve performance further.

## H LLM USAGE STATEMENT

In this work, LLMs are utilized as general-purpose assist tools for programming and writing. Specifically, LLMs assist in code generation and debugging, checking for grammatical errors, and refining the language of the manuscript. No novel research ideas, analyses, or conclusions are contributed by LLMs.

## I EXTENDED PERFORMANCE COMPARISONS

This section provides additional results comparing model performance across six benchmarks under alternative settings, as shown in Tables 10 to 14. We include detailed comparisons among different methods to support our findings in the main text. The results further demonstrate that SASFT effectively maintains model capabilities while reducing code-switching, and in several cases, achieves improved performance relative to SFT. These additional experiments validate the robustness and consistency of our conclusions.

Table 10: Performance comparison on six benchmarks across different methods. We evaluate models trained on the Korean 110k dataset setting. The red numbers indicate performance improvements compared to the SFT.

| Model | Method | MMLU | HumanEval | Flores | HellaSwag | LogiQA | IFEval | MGSM |
|---|---|---|---|---|---|---|---|---|
| | | Acc (%) | Acc (%) | Bleu (%) | Acc (%) | Acc (%) | Acc (%) | Acc (%) |
| Gemma-2-2B | SFT | 27.56 | 77.60 | 18.39 | 26.43 | 26.00 | 15.85 | 11.89 |
| | SFT+GRPO | 27.82 (+0.26) | 76.25 (-1.35) | 18.49 (+0.10) | 21.34 (-5.09) | 26.87 (+0.87) | 16.08 (+0.23) | 12.00 (+0.11) |
| | SFT+Penalty | 26.77 (-0.79) | 77.07 (-0.53) | 18.48 (+0.09) | 22.12 (-4.31) | 26.25 (+0.25) | 16.26 (+0.41) | 12.77 (+0.88) |
| | SASFT | 26.68 (-0.88) | 75.29 (-2.31) | 17.96 (-0.43) | 22.01 (-4.42) | 26.25 (+0.25) | 15.81 (-0.04) | 11.31 (-0.58) |
| Gemma-2-9B | SFT | 47.56 | 96.63 | 29.23 | 34.52 | 30.87 | 24.54 | 48.67 |
| | SFT+GRPO | 47.47 (-0.09) | 96.35 (-0.28) | 29.68 (+0.45) | 33.33 (-1.19) | 33.75 (+2.88) | 24.12 (-0.42) | 50.88 (+2.21) |
| | SFT+Penalty | 46.66 (-0.90) | 95.96 (-0.67) | 29.18 (-0.05) | 34.38 (-0.14) | 29.25 (-1.62) | 25.14 (+0.60) | 46.37 (-2.30) |
| | SASFT | 46.85 (-0.71) | 94.62 (-2.01) | 28.19 (-1.04) | 33.60 (-0.92) | 29.12 (-1.75) | 25.24 (+0.70) | 47.12 (-1.55) |
| Llama-3.1-8B | SFT | 32.07 | 90.14 | 24.73 | 27.60 | 32.25 | 21.99 | 15.92 |
| | SFT+GRPO | 27.73 (-4.34) | 77.16 (-12.98) | 20.75 (-3.98) | 25.68 (-1.92) | 30.63 (-1.62) | 17.75 (-4.24) | 9.44 (-6.48) |
| | SFT+Penalty | 32.30 (+0.23) | 89.18 (-0.96) | 24.68 (-0.05) | 29.20 (+1.60) | 30.63 (-1.62) | 22.03 (+0.04) | 18.37 (+2.45) |
| | SASFT | 32.40 (+0.33) | 87.55 (-2.59) | 24.05 (-0.68) | 30.20 (+2.60) | 32.00 (-0.25) | 20.96 (-1.03) | 13.49 (-2.43) |
| Qwen3-1.7B-Base | SFT | 38.07 | 85.91 | 22.49 | 32.50 | 31.00 | 18.98 | 32.03 |
| | SFT+GRPO | 37.47 (-0.60) | 88.32 (+2.41) | 23.04 (+0.55) | 34.14 (+1.64) | 31.62 (+0.62) | 19.31 (+0.33) | 32.03 (0.00) |
| | SFT+Penalty | 37.94 (-0.13) | 87.02 (+1.11) | 22.58 (+0.09) | 33.53 (+1.03) | 34.38 (+3.38) | 19.17 (+0.19) | 33.31 (+1.28) |
| | SASFT | 37.49 (-0.58) | 86.39 (+0.48) | 22.97 (+0.48) | 34.15 (+1.65) | 34.25 (+3.25) | 19.12 (+0.14) | 32.88 (+0.85) |
| Qwen3-8B-Base | SFT | 49.67 | 97.74 | 22.86 | 34.52 | 39.00 | 35.92 | 59.47 |
| | SFT+GRPO | 45.27 (-4.40) | 96.35 (-1.39) | 24.81 (+1.95) | 22.53 (-11.99) | 39.12 (+0.12) | 34.47 (-1.45) | 55.23 (-4.24) |
| | SFT+Penalty | 47.68 (-1.99) | 95.10 (-2.64) | 26.12 (+3.26) | 30.33 (-4.19) | 38.12 (-0.88) | 35.52 (-0.40) | 60.72 (+1.25) |
| | SASFT | 52.88 (+3.21) | 94.90 (-2.84) | 18.96 (-3.90) | 39.20 (+4.68) | 41.50 (+2.50) | 34.89 (-1.03) | 61.92 (+2.45) |

Table 11: Performance comparison on six benchmarks across different methods. We evaluate models trained on the Korean 210k dataset setting. The red numbers indicate performance improvements compared to SFT.

| Model | Method | MMLU | HumanEval | Flores | HellaSwag | LogiQA | IFEval | MGSM |
|---|---|---|---|---|---|---|---|---|
| | | Acc (%) | Acc (%) | Bleu (%) | Acc (%) | Acc (%) | Acc (%) | Acc (%) |
| Gemma-2-2B | SFT | 25.96 | 75.87 | 19.31 | 19.97 | 24.62 | 16.24 | 14.00 |
| | SFT+GRPO | 25.98 (+0.02) | 78.22 (+2.35) | 19.35 (+0.04) | 19.55 (-0.42) | 25.25 (+0.63) | 16.27 (+0.03) | 13.63 (-0.37) |
| | SFT+Penalty | 26.58 (+0.62) | 79.76 (+3.89) | 15.45 (-3.86) | 22.09 (+2.12) | 29.12 (+4.50) | 16.66 (+0.42) | 13.76 (-0.24) |
| | SASFT | 27.17 (+1.21) | 76.30 (+0.43) | 18.34 (-0.97) | 22.08 (+2.11) | 25.25 (+0.63) | 16.43 (+0.19) | 14.08 (+0.08) |
| Gemma-2-9B | SFT | 50.14 | 92.02 | 29.15 | 42.09 | 33.88 | 23.89 | 49.60 |
| | SFT+GRPO | 49.21 (-0.93) | 91.54 (-0.48) | 28.68 (-0.47) | 42.31 (+0.22) | 32.12 (-1.76) | 23.69 (-0.20) | 53.44 (+3.84) |
| | SFT+Penalty | 50.38 (+0.24) | 93.22 (+1.20) | 29.29 (+0.14) | 47.55 (+5.46) | 30.50 (-3.38) | 23.89 (0.00) | 50.43 (+0.83) |
| | SASFT | 49.33 (-0.81) | 92.69 (+0.67) | 28.87 (-0.28) | 40.13 (-1.96) | 34.75 (+0.87) | 23.60 (-0.29) | 50.88 (+1.28) |
| Llama-3.1-8B | SFT | 34.98 | 89.57 | 23.68 | 33.72 | 28.50 | 22.29 | 22.67 |
| | SFT+GRPO | 35.15 (+0.17) | 89.23 (-0.34) | 23.79 (+0.11) | 31.38 (-2.34) | 31.00 (+2.50) | 22.61 (+0.32) | 22.69 (+0.02) |
| | SFT+Penalty | 35.26 (+0.28) | 88.85 (-0.72) | 23.44 (-0.24) | 35.06 (+1.34) | 30.75 (+2.25) | 22.51 (+0.22) | 27.44 (+4.77) |
| | SASFT | 35.27 (+0.29) | 86.83 (-2.74) | 23.25 (-0.43) | 33.01 (-0.71) | 33.50 (+5.00) | 22.03 (-0.26) | 25.09 (+2.42) |
| Qwen3-1.7B-Base | SFT | 37.02 | 85.10 | 22.40 | 31.73 | 31.25 | 20.19 | 36.69 |
| | SFT+GRPO | 36.96 (-0.06) | 85.19 (+0.09) | 22.44 (+0.04) | 34.47 (+2.74) | 31.87 (+0.62) | 20.07 (-0.12) | 36.72 (+0.03) |
| | SFT+Penalty | 36.57 (-0.45) | 84.13 (-0.97) | 22.50 (+0.10) | 35.41 (+3.68) | 33.00 (+1.75) | 21.01 (+0.82) | 36.93 (+0.24) |
| | SASFT | 37.36 (+0.34) | 85.19 (+0.09) | 22.55 (+0.15) | 31.17 (-0.56) | 31.00 (-0.25) | 20.14 (-0.05) | 37.12 (+0.43) |
| Qwen3-8B-Base | SFT | 49.64 | 96.88 | 26.37 | 37.40 | 39.38 | 34.78 | 65.71 |
| | SFT+GRPO | 48.19 (-1.45) | 97.74 (+0.86) | 27.07 (+0.70) | 34.87 (-2.53) | 41.38 (+2.00) | 34.18 (-0.60) | 61.87 (-3.84) |
| | SFT+Penalty | 50.80 (+1.16) | 96.15 (-0.73) | 24.83 (-1.54) | 40.32 (+2.92) | 40.50 (+1.12) | 35.94 (+1.16) | 64.13 (-1.58) |
| | SASFT | 51.36 (+1.72) | 95.77 (-1.11) | 21.80 (-4.57) | 45.68 (+8.28) | 42.88 (+3.50) | 35.27 (+0.49) | 63.92 (-1.79) |

Table 12: Performance comparison on six benchmarks across different methods. We evaluate models trained on the Russian 110k dataset setting. The red numbers indicate performance improvements compared to SFT.

| Model | Method | MMLU | HumanEval | Flores | HellaSwag | LogiQA | IFEval | MGSM |
|---|---|---|---|---|---|---|---|---|
| | | Acc (%) | Acc (%) | Bleu (%) | Acc (%) | Acc (%) | Acc (%) | Acc (%) |
| Gemma-2-2B | SFT | 24.74 | 82.45 | 23.25 | 17.35 | 24.87 | 16.65 | 11.71 |
| | SFT+GRPO | 25.14 (+0.40) | 83.85 (+1.40) | 23.54 (+0.29) | 14.58 (-2.77) | 27.25 (+2.38) | 16.81 (+0.16) | 10.80 (-0.91) |
| | SFT+Penalty | 26.77 (+2.03) | 85.10 (+2.65) | 22.18 (-1.07) | 19.65 (+2.30) | 29.87 (+5.00) | 16.81 (+0.16) | 12.11 (+0.40) |
| | SASFT | 26.01 (+1.27) | 80.96 (-1.49) | 23.31 (+0.06) | 19.24 (+1.89) | 25.50 (+0.63) | 16.26 (-0.39) | 10.96 (-0.75) |
| Gemma-2-9B | SFT | 42.97 | 94.23 | 31.82 | 33.84 | 33.38 | 23.62 | 44.48 |
| | SFT+GRPO | 42.92 (-0.05) | 93.89 (-0.34) | 31.55 (-0.27) | 36.08 (+2.24) | 31.75 (-1.63) | 23.37 (-0.25) | 43.63 (-0.85) |
| | SFT+Penalty | 42.18 (-0.79) | 96.44 (+2.21) | 30.32 (-1.50) | 32.08 (-1.76) | 29.88 (-3.50) | 21.76 (-1.86) | 41.52 (-2.96) |
| | SASFT | 40.76 (-2.21) | 96.68 (+2.45) | 31.31 (-0.51) | 29.86 (-3.98) | 31.87 (-1.51) | 22.23 (-1.39) | 44.40 (-0.08) |
| Llama-3.1-8B | SFT | 29.96 | 92.40 | 21.45 | 23.71 | 29.38 | 19.59 | 15.76 |
| | SFT+GRPO | 29.84 (-0.12) | 91.49 (-0.91) | 21.82 (+0.37) | 21.80 (-1.91) | 29.62 (+0.24) | 19.19 (-0.40) | 15.15 (-0.61) |
| | SFT+Penalty | 33.88 (+3.92) | 89.23 (-3.17) | 25.49 (+4.04) | 30.44 (+6.73) | 29.75 (+0.37) | 20.24 (+0.65) | 17.49 (+1.73) |
| | SASFT | 32.06 (+2.10) | 92.98 (+0.58) | 23.52 (+2.07) | 29.53 (+5.82) | 32.88 (+3.50) | 20.37 (+0.78) | 17.44 (+1.68) |
| Qwen3-1.7B-Base | SFT | 37.22 | 90.00 | 23.46 | 35.53 | 32.25 | 19.88 | 33.25 |
| | SFT+GRPO | 37.77 (+0.55) | 90.72 (+0.72) | 23.84 (+0.38) | 34.80 (-0.73) | 29.75 (-2.50) | 20.26 (+0.38) | 32.69 (-0.56) |
| | SFT+Penalty | 37.47 (+0.25) | 90.05 (+0.05) | 23.68 (+0.22) | 32.79 (-2.74) | 29.88 (-0.62) | 20.64 (+0.76) | 33.47 (+0.22) |
| | SASFT | 38.20 (+0.98) | 91.11 (+1.11) | 24.56 (+1.10) | 34.92 (-0.61) | 33.62 (+1.37) | 19.94 (+0.06) | 32.43 (-0.82) |
| Qwen3-8B-Base | SFT | 47.21 | 94.13 | 25.77 | 35.42 | 41.38 | 30.92 | 50.03 |
| | SFT+GRPO | 45.04 (-2.17) | 94.33 (+0.20) | 26.86 (+1.09) | 28.03 (-7.39) | 40.62 (-0.76) | 29.62 (-1.30) | 48.75 (-1.28) |
| | SFT+Penalty | 45.73 (-1.48) | 95.00 (+0.87) | 26.89 (+1.12) | 28.35 (-7.07) | 41.00 (-0.38) | 30.80 (-0.12) | 50.03 (0.00) |
| | SASFT | 50.28 (+3.07) | 88.89 (-5.24) | 26.95 (+1.18) | 38.47 (+3.05) | 44.50 (+3.12) | 32.35 (+1.43) | 53.89 (+3.86) |

Table 13: Performance comparison on six benchmarks across different methods. We evaluate models trained on the Russian 210k dataset setting. The red numbers indicate performance improvements compared to SFT.

| Model | Method | MMLU | HumanEval | Flores | HellaSwag | LogiQA | IFEval | MGSM |
|---|---|---|---|---|---|---|---|---|
| | | Acc (%) | Acc (%) | Bleu (%) | Acc (%) | Acc (%) | Acc (%) | Acc (%) |
| Gemma-2-2B | SFT | 28.36 | 87.69 | 23.19 | 24.84 | 31.50 | 17.22 | 14.83 |
| | SFT+GRPO | 28.04 (-0.32) | 88.65 (+0.96) | 23.35 (+0.16) | 25.63 (+0.79) | 29.38 (-2.12) | 17.17 (-0.05) | 14.08 (-0.75) |
| | SFT+Penalty | 28.32 (-0.04) | 86.88 (-0.81) | 23.06 (-0.13) | 25.34 (+0.50) | 27.00 (-4.50) | 17.08 (-0.14) | 13.84 (-0.99) |
| | SASFT | 28.09 (-0.27) | 88.46 (+0.77) | 23.25 (+0.06) | 26.67 (+1.83) | 26.75 (-4.75) | 16.44 (-0.78) | 13.44 (-1.39) |
| Gemma-2-9B | SFT | 45.55 | 96.78 | 30.51 | 35.44 | 33.75 | 22.82 | 50.05 |
| | SFT+GRPO | 44.99 (-0.56) | 96.92 (+0.14) | 30.65 (+0.14) | 36.64 (+1.20) | 33.25 (-0.50) | 22.76 (-0.06) | 50.75 (+0.70) |
| | SFT+Penalty | 44.51 (-1.04) | 96.83 (+0.05) | 30.67 (+0.16) | 36.11 (+0.67) | 35.00 (+1.25) | 23.18 (+0.36) | 50.56 (+0.51) |
| | SASFT | 43.55 (-2.00) | 94.81 (-1.97) | 22.93 (-7.58) | 32.71 (-2.73) | 31.87 (-1.88) | 21.83 (-0.99) | 49.79 (-0.26) |
| Llama-3.1-8B | SFT | 33.97 | 93.94 | 23.24 | 29.72 | 30.25 | 21.24 | 14.51 |
| | SFT+GRPO | 33.71 (-0.26) | 94.37 (+0.43) | 23.47 (+0.23) | 31.76 (+2.04) | 28.38 (-1.87) | 20.92 (-0.32) | 14.35 (-0.16) |
| | SFT+Penalty | 33.29 (-0.68) | 96.39 (+2.45) | 24.00 (+0.76) | 31.31 (+1.59) | 32.00 (+1.75) | 22.24 (+1.00) | 13.25 (-1.26) |
| | SASFT | 34.53 (+0.56) | 96.59 (+2.65) | 23.24 (0.00) | 29.87 (+0.15) | 30.75 (+0.50) | 21.70 (+0.46) | 18.88 (+4.37) |
| Qwen3-1.7B-Base | SFT | 38.06 | 93.75 | 23.76 | 33.65 | 31.75 | 20.72 | 35.04 |
| | SFT+GRPO | 37.88 (-0.18) | 92.07 (-1.68) | 23.41 (-0.35) | 35.05 (+1.40) | 31.00 (-0.75) | 20.89 (+0.17) | 34.61 (-0.43) |
| | SFT+Penalty | 38.38 (+0.32) | 94.47 (+0.72) | 23.29 (-0.47) | 33.55 (-0.10) | 36.00 (+4.25) | 20.37 (-0.35) | 34.99 (-0.05) |
| | SASFT | 38.23 (+0.17) | 93.12 (-0.63) | 23.14 (-0.62) | 33.96 (+0.31) | 32.38 (+0.63) | 21.14 (+0.42) | 34.53 (-0.51) |
| Qwen3-8B-Base | SFT | 50.73 | 96.44 | 28.31 | 38.99 | 43.12 | 35.08 | 60.27 |
| | SFT+GRPO | 48.63 (-2.10) | 95.14 (-1.30) | 28.40 (+0.09) | 34.01 (-4.98) | 43.12 (0.00) | 33.98 (-1.10) | 57.87 (-2.40) |
| | SFT+Penalty | 51.56 (+0.83) | 95.72 (-0.72) | 28.66 (+0.35) | 40.60 (+1.61) | 42.62 (-0.50) | 34.82 (-0.26) | 55.28 (-4.99) |
| | SASFT | 52.11 (+1.38) | 95.24 (-1.20) | 26.69 (-1.62) | 44.83 (+5.84) | 42.62 (-0.50) | 35.74 (+0.66) | 58.19 (-2.08) |

Table 14: Performance comparison on six benchmarks across different methods. We evaluate models trained on the Chinese 210k dataset setting. The red numbers indicate performance improvements compared to SFT.

| Model | Method | MMLU | HumanEval | Flores | HellaSwag | LogiQA | IFEval | MGSM |
|-------|--------|------|-----------|--------|-----------|--------|--------|------|
| | | Acc (%) | Acc (%) | Bleu (%) | Acc (%) | Acc (%) | Acc (%) | Acc (%) |
| Gemma-2-2B | SFT | 28.58 | 91.25 | 23.68 | 27.47 | 29.50 | 15.65 | 14.61 |
| | SFT+GRPO | 28.99 (+0.41) | 90.87 (-0.38) | 23.25 (-0.43) | 28.50 (+1.03) | 25.75 (-3.75) | 16.14 (+0.49) | 14.80 (+0.19) |
| | SFT+Penalty | 28.80 (+0.22) | 90.77 (-0.48) | 23.42 (-0.26) | 27.85 (+0.38) | 26.00 (-3.50) | 15.94 (+0.29) | 15.44 (+0.83) |
| | SASFT | 27.89 (-0.69) | 90.82 (-0.43) | 22.96 (-0.72) | 28.97 (+1.50) | 28.12 (-1.38) | 15.80 (+0.15) | 14.61 (0.00) |
| Gemma-2-9B | SFT | 45.77 | 93.70 | 29.37 | 33.92 | 31.63 | 24.58 | 49.63 |
| | SFT+GRPO | 46.22 (+0.45) | 94.09 (+0.39) | 29.22 (-0.15) | 36.22 (+2.30) | 29.12 (-2.51) | 24.24 (-0.34) | 48.72 (-0.91) |
| | SFT+Penalty | 45.39 (-0.38) | 91.73 (-1.97) | 29.33 (-0.04) | 34.78 (+0.86) | 32.38 (+0.75) | 23.84 (-0.74) | 48.99 (-0.64) |
| | SASFT | 47.04 (+1.27) | 92.50 (-1.20) | 28.79 (-0.58) | 34.11 (+0.19) | 33.13 (+1.50) | 25.50 (+0.92) | 50.29 (+0.66) |
| Llama-3.1-8B | SFT | 31.53 | 91.35 | 22.70 | 28.88 | 30.00 | 21.28 | 16.13 |
| | SFT+GRPO | 30.35 (-1.18) | 89.33 (-2.02) | 22.42 (-0.28) | 29.93 (+1.05) | 30.62 (+0.62) | 21.22 (-0.06) | 13.65 (-2.48) |
| | SFT+Penalty | 33.37 (+1.84) | 88.51 (-2.84) | 25.09 (+2.39) | 29.79 (+0.91) | 28.62 (-1.38) | 22.32 (+1.04) | 19.23 (+3.10) |
| | SASFT | 33.37 (+1.84) | 95.38 (+4.03) | 24.68 (+1.98) | 33.80 (+4.92) | 31.62 (+1.62) | 23.01 (+1.73) | 20.56 (+4.43) |
| Qwen3-1.7B-Base | SFT | 37.27 | 93.22 | 23.59 | 32.30 | 34.00 | 20.53 | 32.48 |
| | SFT+GRPO | 36.99 (-0.28) | 93.12 (-0.10) | 23.68 (+0.09) | 34.20 (+1.90) | 30.87 (-3.13) | 20.78 (+0.25) | 32.40 (-0.08) |
| | SFT+Penalty | 37.76 (+0.49) | 92.69 (-0.53) | 23.21 (-0.38) | 35.66 (+3.36) | 31.38 (-2.62) | 21.07 (+0.54) | 34.51 (+2.03) |
| | SASFT | 38.10 (+0.83) | 92.12 (-1.10) | 23.56 (-0.03) | 34.20 (+1.90) | 33.50 (-0.50) | 20.93 (+0.40) | 33.01 (+0.53) |
| Qwen3-8B-Base | SFT | 49.53 | 96.83 | 30.20 | 31.58 | 42.50 | 34.67 | 55.09 |
| | SFT+GRPO | 44.85 (-4.68) | 96.30 (-0.53) | 30.81 (+0.61) | 24.72 (-6.86) | 42.12 (-0.38) | 33.73 (-0.94) | 50.75 (-4.34) |
| | SFT+Penalty | 48.64 (-0.89) | 96.83 (0.00) | 30.75 (+0.55) | 33.42 (+1.84) | 40.88 (-1.62) | 35.66 (+0.99) | 57.25 (+2.16) |
| | SASFT | 49.60 (+0.07) | 96.92 (+0.09) | 30.81 (+0.61) | 37.18 (+5.60) | 43.38 (+0.88) | 33.90 (-0.77) | 51.95 (-3.14) |

## J ROBUSTNESS OF LANGUAGE-SPECIFIC FEATURES ACROSS SAE CONFIGURATIONS

To investigate the robustness of language-specific features to different SAE hyperparameters, we conduct experiments using SAEs with varying sparsity (l0) and dimensionality (width) from Gemma Scope (Lieberum et al., 2024). Specifically, we examine six different SAE settings: l0_38 width_16k, l0_34 width_65k, l0_73 width_16k, l0_63 width_65k, l0_158 width_16k, and l0_124 width_65k.

For each SAE configuration, we identify the rank #0 language-specific feature for Chinese and Korean using the method described in Section 4.1. We then compute the pairwise cosine similarity between these features across different SAE configurations for layers 19 through 25. The results are visualized as heatmaps in Figures 8-21.

Our findings demonstrate that language-specific features exhibit remarkable consistency across different SAE hyperparameters. For both Chinese and Korean, the cosine similarities between rank #0 features from different SAE configurations typically exceed 0.85, with many similarities above 0.90. This high degree of similarity persists across all examined layers (19-25), indicating that:

- Language-specific features are robust to variations in SAE sparsity targets (l0 values ranging from 34 to 158)
- Feature identification is stable across different SAE dimensionalities (16k vs. 65k width)
- The consistent patterns across multiple layers suggest that language features are fundamental properties captured by SAEs regardless of specific training configurations

These results provide strong evidence that our language feature identification method is reliable and that SASFT's effectiveness is not critically dependent on specific SAE hyperparameter choices.

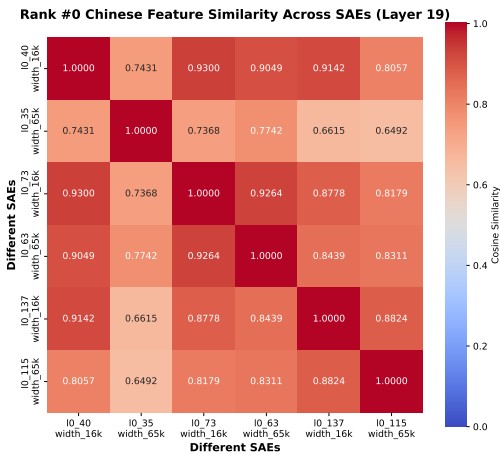

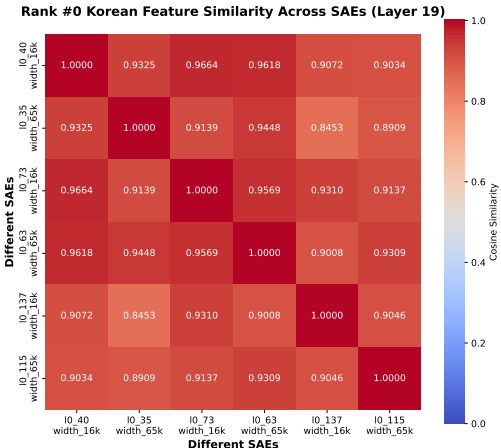

Figure 8: Similarity of rank #0 Chinese features across SAE configurations at layer 19.

Figure 9: Similarity of rank #0 Korean features across SAE configurations at layer 19.

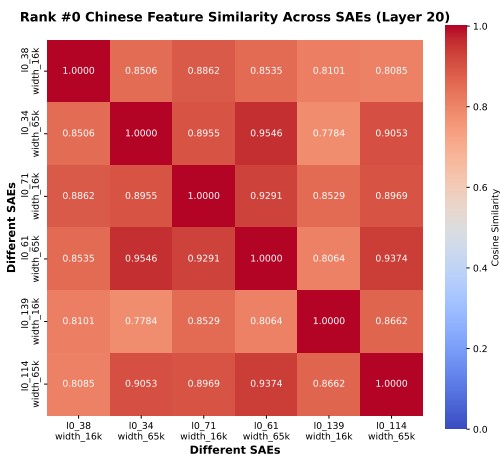

Figure 10: Similarity of rank #0 Chinese features across SAE configurations at layer 20.

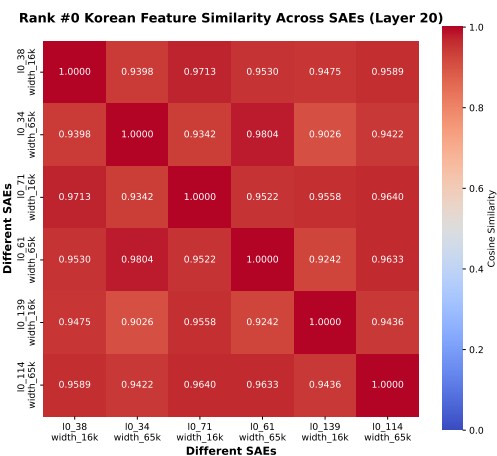

Figure 11: Similarity of rank #0 Korean features across SAE configurations at layer 20.

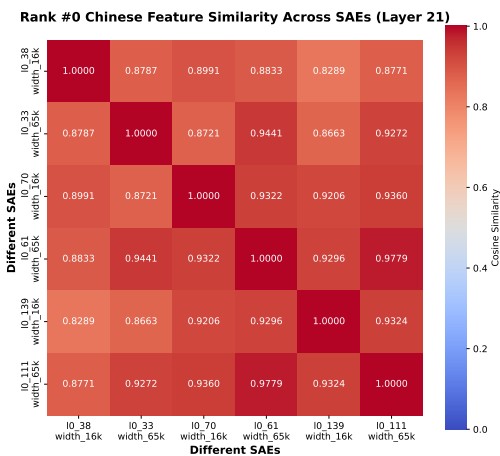

Figure 12: Similarity of rank #0 Chinese features across SAE configurations at layer 21.

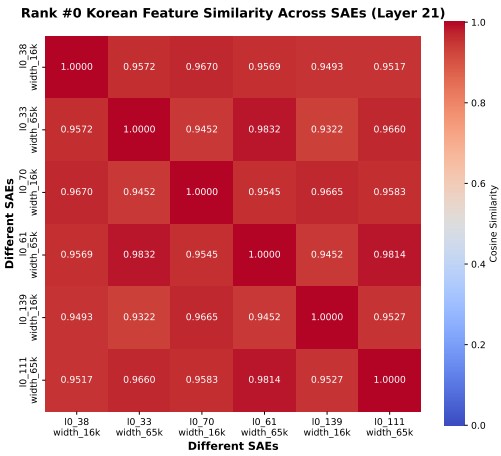

Figure 13: Similarity of rank #0 Korean features across SAE configurations at layer 21.

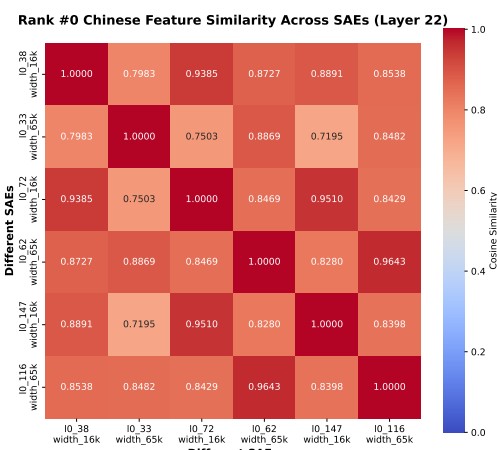

Figure 14: Similarity of rank #0 Chinese features across SAE configurations at layer 22.

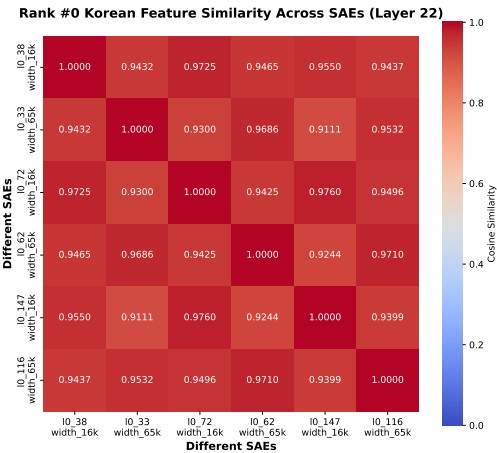

Figure 15: Similarity of rank #0 Korean features across SAE configurations at layer 22.

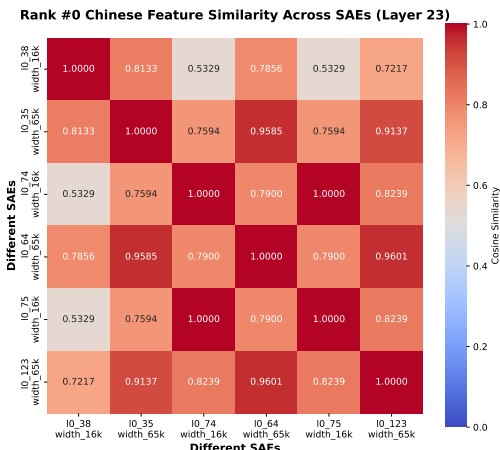

Figure 16: Similarity of rank #0 Chinese features across SAE configurations at layer 23.

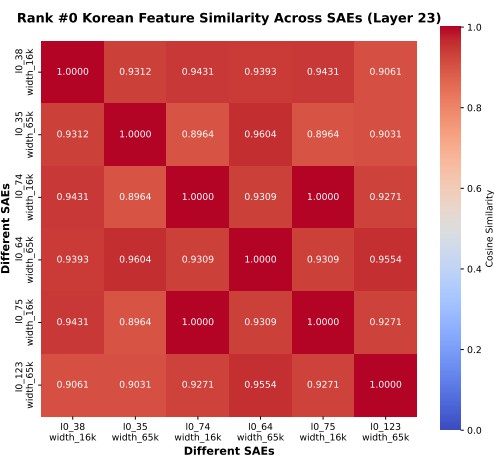

Figure 17: Similarity of rank #0 Korean features across SAE configurations at layer 23.

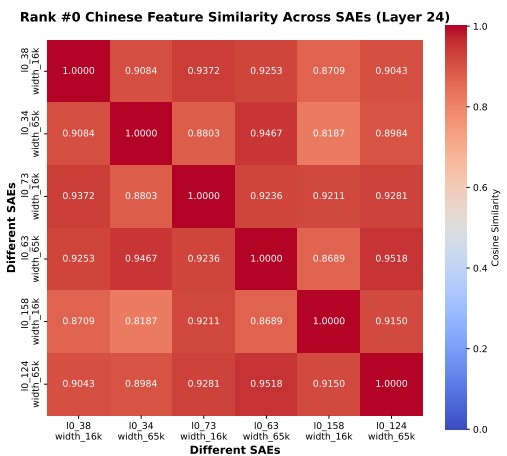

Figure 18: Similarity of rank #0 Chinese features across SAE configurations at layer 24.

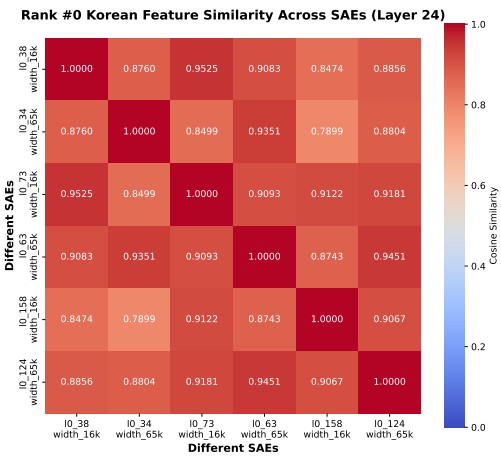

Figure 19: Similarity of rank #0 Korean features across SAE configurations at layer 24.

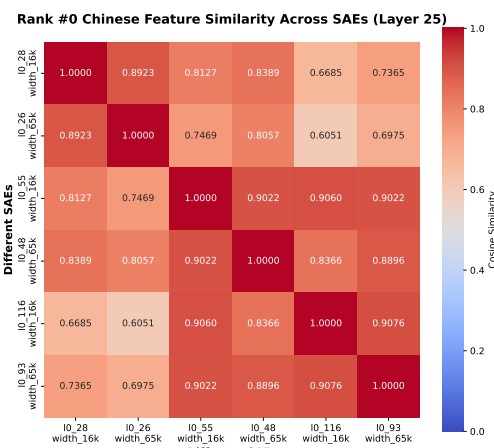

Figure 20: Similarity of rank #0 Chinese features across SAE configurations at layer 25.

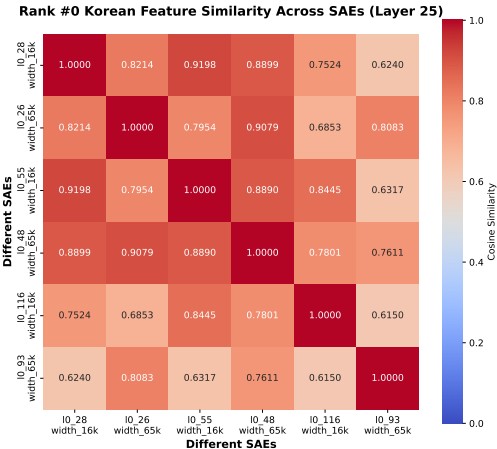

Figure 21: Similarity of rank #0 Korean features across SAE configurations at layer 25.

# K CAUSAL EVIDENCE: ENHANCING LANGUAGE FEATURES INDUCES CODE-SWITCHING

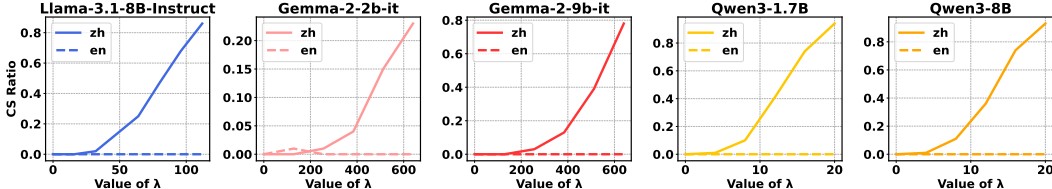

Figure 22: Code-switching ratio to Chinese after enhancing Chinese or English features with different $\lambda$ values. (1) Enhancing the Chinese feature can induce unexpected code-switching. (2) A higher coefficient $\lambda$ leads to higher code-switching ratio. (3) Enhancing the English feature has little impact on the code-switching ratio to Chinese.

To establish a causal relationship between language-specific feature activation and code-switching, we conduct the inverse experiment of Section 3.3.2. While ablation demonstrates that reducing language feature activation decreases code-switching, we now test whether artificially increasing the activation of a target language feature can induce code-switching. Specifically, we use *directional enhancement* to add the language feature to the residual stream $\mathbf{x} \in \mathbb{R}^N$ at the final layer of a randomly selected token. This process can be expressed as:

$$\mathbf{x}' \leftarrow \mathbf{x} + \lambda \mathbf{d}, \tag{12}$$

where $\mathbf{d}$ represents the language feature and $\lambda$ is the coefficient that controls the degree of enhancement. After obtaining $\mathbf{x}'$, we replace $\mathbf{x}$ with $\mathbf{x}'$ and continue the forward pass of the LLMs. We test this on 100 samples that originally contained no code-switching to Chinese and report the code-switching ratio to Chinsese with different $\lambda$ in Figure 22. Our observations are as follows: (1) Enhancing the Chinese feature induces unexpected code-switching across all models. (2) A higher coefficient $\lambda$ leads to higher code-switching ratios. (3) Enhancing English features has minimal impact on code-switching behavior. These results, combined with our ablation experiments, provide bidirectional causal evidence: artificially manipulating language-specific feature activations can both induce and suppress code-switching behavior, strongly supporting our hypothesis that language-specific feature activation causally determines language selection in LLM generation.

