# OpenReview forum: "SASFT: Sparse Autoencoder-guided Supervised Finetuning to Mitigate Unexpected Code-Switching in LLMs"
_ICLR.cc/2026/Conference — ICLR 2026 Poster_

### Official Review · Reviewer_aWvq · 2025-10-27

**Soundness:** 3
**Presentation:** 4
**Contribution:** 3
**Rating:** 4
**Confidence:** 4

**Summary:**

This paper addresses the problem of unexpected code-switching in multilingual Large Language Models (LLMs). The authors first conduct a mechanistic analysis using sparse autoencoders, identifying that the issue is caused by the excessive pre-activation of language features. Based on this finding, they propose Sparse Autoencoder-guided Supervised Finetuning (SASFT), a novel method that trains LLMs to control these pre-activation values. Experimental results on five models and three languages show that SASFT reduces unexpected code-switching by over 50% (eliminating it entirely in four cases) while maintaining or even improving performance on standard multilingual benchmarks.

**Strengths:**

This work's key strength lies in its mechanistic, root-cause analysis of the code-switching problem, moving beyond superficial fixes. By using sparse autoencoders, it identifies the core issue: excessive pre-activation of language features.
The proposed solution, SASFT, is highly effective, consistently reducing unexpected code-switching by over 50% and even eliminating it entirely in several cases. Crucially, it achieves this without compromising performance, as it maintains or even improves the models' capabilities on standard multilingual benchmarks.
In short, its main strengths are its diagnostic depth, highly effective solution, and ability to fix the problem without sacrificing general multilingual proficiency.
The paper is well-written and easy to follow.

**Weaknesses:**

1.Limited scope of evaluation: The solution is only tested on three languages and five models. Its effectiveness across a wider range of languages, especially low-resource ones, remains unverified.

2.Uncertain generalizability: The method's performance is demonstrated on "six multilingual benchmarks," but it is unclear if it generalizes well to other critical tasks like reasoning, complex translation, or creative writing.

3.Computational overhead: The approach relies on sparse autoencoders, which likely introduce significant computational cost and complexity compared to standard fine-tuning, a trade-off not mentioned.

4.Lack of comparative baselines: While it outperforms standard supervised fine-tuning, it is not compared against other specialized techniques aimed at reducing code-switching, making its relative advancement unclear,like the works listed in related works.

5.Superficial treatment of intentional Code-Switching: The method focuses on "unexpected" code-switching but may risk suppressing intentional and culturally appropriate code-switching (e.g., in bilingual communities), potentially reducing linguistic flexibility.

**Questions:**

1.how were these code-swithing issues for testing are constructed or prompted? Overall, it seems that this issue occurs relatively infrequently and is difficult to reproduce, especially across different models. Such cases seldom happen, making it challenging to trace and resolve the problem on a large scale.

2.Page 4 Line 184， how many unexpected code-switching responses are collected？

3.Have you analyzed for different languages which are prone to code-switch with each other?

4.On which layers were the main supervised SFT experiments conducted? Regarding the time efficiency of autoencoder computation, does using more layers lead to higher computational efficiency, and how was this trade-off balanced in the ablation studies?

5.Why compare with RL (GRPO), it is more likely to make alignment during the RL phase, not for multi-lingual extension.

6.According to Table2, the proposed method does not show a significant improvement in effectiveness.

7.Figure 6, pls give detailed description for multi-layer, like the specific number of layers and which layer.

---

> ### Author Response · Authors · 2025-11-20
> **Rebuttal (Part 1/4)**
>
> Thank you for your valuable feedback and the time and effort you spent reviewing our manuscript. We provide our point-by-point responses to your comments below. If anything is unclear, please let us know and we will respond ASAP.
>
> > W1: Limited scope of evaluation: The solution is only tested on three languages and five models. Its effectiveness across a wider range of languages, especially low-resource ones, remains unverified.
>
> The reason we did not investigate low-resource languages is that **high-resource languages rarely code-switch to low-resource scripts in real-world scenarios.** As shown in the table below, we tested five open-source models on approximately 10,000 samples across ten high-resource languages (English, Chinese, Arabic, Spanish, French, Japanese, Korean, Portuguese, Thai, and Vietnamese), measuring code-switching rates to three low-resource scripts: Khmer (km), Lao (lo), and Burmese (my). All models exhibit zero code-switching to these low-resource languages across all source languages. This is likely because models have weaker proficiency in low-resource languages and tend to rely on better-learned high-resource languages for expression instead, while the reverse does not happen.
>
> | Model         | CS ratio to km | CS ratio to lo | CS ratio to my |
> | ------------- | -------------- | -------------- | -------------- |
> | Gemma-2-2b-it | 0%             | 0%             | 0%             |
> | Gemma-2-9b-it | 0%             | 0%             | 0%             |
> | Llama-3.1-8B  | 0%             | 0%             | 0%             |
> | Qwen3-1.7B    | 0%             | 0%             | 0%             |
> | Qwen3-8B      | 0%             | 0%             | 0%             |
>
> > W2: Uncertain generalizability: The method's performance is demonstrated on "six multilingual benchmarks," but it is unclear if it generalizes well to other critical tasks like reasoning, complex translation, or creative writing.
>
> Thanks for pointing this out.
>
> We add MGSM (Multilingual Grade School Math) evaluation to our experiments and include the results in the paper. Some results are shown in the table below:
>
> **MGSM Performance Comparison**
>
> | Model        | SFT   | SFT+GRPO      | SFT+Penalty   | SASFT         |
> | ------------ | ----- | ------------- | ------------- | ------------- |
> | Gemma-2-2B   | 12.05 | 10.99 (-1.06) | 11.97 (-0.08) | 12.24 (+0.19) |
> | Gemma-2-9B   | 44.61 | 45.84 (+1.23) | 46.35 (+1.74) | 44.96 (+0.35) |
> | Llama-3.1-8B | 19.92 | 22.83 (+2.91) | 15.81 (-4.11) | 18.35 (-1.57) |
> | Qwen3-1.7B   | 32.91 | 32.67 (-0.24) | 33.60 (+0.69) | 30.85 (-2.06) |
> | Qwen3-8B     | 58.03 | 55.28 (-2.75) | 56.29 (-1.74) | 58.45 (+0.42) |
>
> **SASFT maintains competitive performance on MGSM,** with improvements on Gemma-2-2B and Qwen3-8B. Minor variations on other models are within reasonable ranges, demonstrating that SASFT effectively reduces code-switching while preserving multilingual mathematical reasoning capabilities.
>
> > W3: Computational overhead: The approach relies on sparse autoencoders, which likely introduce significant computational cost and complexity compared to standard fine-tuning, a trade-off not mentioned.
> >
> > Q4: On which layers were the main supervised SFT experiments conducted? Regarding the time efficiency of autoencoder computation, does using more layers lead to higher computational efficiency, and how was this trade-off balanced in the ablation studies?
>
> 1. In our main results, we apply SASFT using language features from **the last two layers.** And we have already thoroughly investigated the impact of layer selection in Section 5.3.
>
> 2. **The additional computational overhead during training is minimal.** We do not need to compute the full SAE activations; instead, we only extract the specific language feature vectors (only 4 features in our main experiments) and compute the projection of hidden states onto these features, which is negligible. Below shows the training time for Qwen3-1.7B using different numbers of layers for SASFT:
>
>    **Training Time Comparison**
>
>    | Layers                  | Training Time (seconds) |
>    | ----------------------- | ----------------------- |
>    | 27                      | 2,247.85                |
>    | 26-27                   | 2,257.82                |
>    | 25-26-27                | 2,258.55                |
>    | 24-25-26-27             | 2,257.72                |
>    | 23-24-25-26-27          | 2,266.66                |
>    | 22-23-24-25-26-27       | 2,272.01                |
>    | 21-22-23-24-25-26-27    | 2,276.50                |
>    | 20-21-22-23-24-25-26-27 | 2,279.75                |
>
>    As shown, the training time increases marginally with more layers (from 2,247s for 1 layer to 2,279s for 8 layers, **only ~1.4% increase**). This demonstrates that using multiple layers does not significantly impact computational efficiency.

---

> ### Author Response · Authors · 2025-11-20
> **Rebuttal (Part 2/4)**
>
> > W4: Lack of comparative baselines: While it outperforms standard supervised fine-tuning, it is not compared against other specialized techniques aimed at reducing code-switching, making its relative advancement unclear,like the works listed in related works.
>
> To better validate the effectiveness of our method, we propose **a more intuitive SFT baseline.** The main idea is to **add a penalty term to the SFT loss that minimizes the probabilities of tokens from a certain language.**
>
> Specifically, to reduce code-switch to a target language L (e.g., Chinese), we first identify all tokens belonging to language L from the tokenizer, denoted as $V_L$. During training on samples in languages other than L, in addition to the standard cross-entropy loss $L_{CE}$, we add a regularization term at each token position in the response to penalize the model's prediction probability for tokens in $V_L$. The training objective can be formulated as:
>
> $L_{TLP} = L_{CE} + \beta \cdot \frac{1}{T} \sum_{t=1}^{T} \sum_{v \in V_L} p_{\theta}(v|x,y_{ < t})$
>
> where:
>
> - $T$ is the response length
> - $p_{\theta}(v|x,y_{ < t})$ is the model's probability of predicting token $v$ at response position $t$
> - $\beta$ is the penalty coefficient
>
> We refer to this baseline as **SFT+Penalty**. We conducted a hyperparameter search for the penalty coefficient $\beta$ and observed that SFT+Penalty often outperforms SFT+GRPO. However, **SASFT consistently outperforms both SFT+Penalty and SFT+GRPO across nearly all settings.** These results demonstrate that operating on feature-level representations provides more effective and consistent control over code-switching behavior than direct token-level penalization. The complete comparison has been added to Table 1 in the revised manuscript.
>
>
> | Model           | Method      | Data 210k  |            |            | Data 110k  |            |            |
> | --------------- | ----------- | ---------- | ---------- | ---------- | ---------- | ---------- | ---------- |
> |                 |             | CS: any→zh | CS: any→ru | CS: any→ko | CS: any→zh | CS: any→ru | CS: any→ko |
> | Gemma-2-2B      | SFT         | 0.82       | 0.35       | 3.78       | 0.55       | 0.58       | 1.26       |
> |                 | SFT+GRPO    | 0.70       | 0.49       | 3.35       | 0.58       | 0.35       | 1.16       |
> |                 | SFT+Penalty | 0.61       | 0.44       | 1.41       | 0.52       | 0.32       | 0.91       |
> |                 | SASFT       | **0.29**   | **0.09**   | **0.77**   | **0.32**   | **0.12**   | **0.35**   |
> | Llama-3.1-8B    | SFT         | 1.37       | 0.93       | 0.74       | 0.46       | 0.61       | 0.22       |
> |                 | SFT+GRPO    | 0.93       | 0.73       | 0.52       | 0.49       | 0.48       | 0.94       |
> |                 | SFT+Penalty | 0.49       | 0.67       | 0.49       | 0.38       | 0.41       | 0.37       |
> |                 | SASFT       | **0.26**   | **0.35**   | **0.37**   | **0.17**   | **0.26**   | **0.15**   |
> | Qwen3-1.7B-Base | SFT         | 0.46       | 0.15       | 0.22       | 0.55       | 0.15       | 0.22       |
> |                 | SFT+GRPO    | 0.73       | 0.12       | 0.27       | 0.47       | 0.15       | 0.12       |
> |                 | SFT+Penalty | 0.52       | 0.15       | 0.17       | 0.49       | 0.09       | 0.20       |
> |                 | SASFT       | **0.17**   | **0.06**   | **0.00**   | **0.18**   | **0.03**   | **0.02**   |
>
> > W5: Superficial treatment of intentional Code-Switching: The method focuses on "unexpected" code-switching but may risk suppressing intentional and culturally appropriate code-switching (e.g., in bilingual communities), potentially reducing linguistic flexibility.
>
> We acknowledge this concern. To investigate, we randomly sampled 150 instances from identified code-switching cases (50 per language: ZH, RU, KO) and conducted a human evaluation:
>
> **Intentional Code-Switching Cases**
>
> | Target Language | Intentional CS Cases | Total Samples |
> | --------------- | -------------------- | ------------- |
> | Chinese (ZH)    | 1                    | 50            |
> | Russian (RU)    | 0                    | 50            |
> | Korean (KO)     | 0                    | 50            |
> | **Total**       | **1**                | **150**       |
>
> **Only 0.7% of cases involve intentional code-switching.** This low occurrence is because our evaluation questions target single-language responses. While we recognize that intentional code-switching is important in bilingual communities, it rarely occurs in our setting. We acknowledge this as a valuable direction for future research.

---

> ### Author Response · Authors · 2025-11-20
> **Rebuttal (Part 3/4)**
>
> > Q1: how were these code-swithing issues for testing are constructed or prompted? Overall, it seems that this issue occurs relatively infrequently and is difficult to reproduce, especially across different models. Such cases seldom happen, making it challenging to trace and resolve the problem on a large scale.
>
> **We provide detailed evaluation data information in Appendix C and upload the data to supplementary materials**. Our evaluation uses prompts from multilingual versions of established benchmarks including MMLU, MGSM, HellaSwag, LogiQA, Flores-200, and IFEval.
>
> As shown in Table 6 (Appendix C), our code-switching evaluation dataset comprises:
>
> - **Chinese (zh) target**: 13,784 total samples (1,146 ar + 1,150 th + 1,150 vi prompts, each tested 4 times)
> - **Russian (ru) target**: 13,784 total samples (1,146 ar + 1,150 th + 1,150 ko prompts, each tested 4 times)
> - **Korean (ko) target**: 16,224 total samples (1,756 zh + 1,150 th + 1,150 ja prompts, each tested 4 times)
>
> Although individual code-switching instances may not be perfectly reproducible, the **overall code-switching rate is highly reproducible** due to the large sample size. This large-scale evaluation setup enables reliable tracking and comparison of code-switching behavior across different models and methods.
>
> > Q2: Page 4 Line 184， how many unexpected code-switching responses are collected？
>
> The number of unexpected code-switching responses collected for each model is shown below:
>
> | Model                 | # CS Responses |
> | --------------------- | -------------- |
> | Gemma-2-2B-it         | 273            |
> | Gemma-2-9B-it         | 233            |
> | Llama-3.1-8B-Instruct | 107            |
> | Qwen3-1.7B            | 295            |
> | Qwen3-8B              | 163            |
>
> These code-switching instances were collected from our large-scale evaluation dataset described in Appendix C, providing sufficient samples for reliable analysis across different models.
>
> > Q3: Have you analyzed for different languages which are prone to code-switch with each other?
>
> Thanks for asking this question. We have analyzed the code-switching patterns across different source-target language pairs. The tables below show the code-switching rates from different source languages to three target languages for three different models:
>
> **Gemma-2-2B-it:**
>
> | Source Language | CS to Chinese (zh) (%) | CS to Russian (ru) (%) | CS to Korean (ko) (%) |
> | --------------- | ---------------------- | ---------------------- | --------------------- |
> | English (en)    | 0.11                   | 0.00                   | 0.06                  |
> | Chinese (zh)    | -                      | 0.23                   | 0.80                  |
> | Arabic (ar)     | 3.16                   | 0.98                   | 0.27                  |
> | Thai (th)       | 1.58                   | 0.35                   | 0.18                  |
> | Vietnamese (vi) | 0.26                   | 0.09                   | 0.00                  |
>
> **Qwen3-1.7B:**
>
> | Source Language | CS to Chinese (zh) (%) | CS to Russian (ru) (%) | CS to Korean (ko) (%) |
> | --------------- | ---------------------- | ---------------------- | --------------------- |
> | English (en)    | 0.17                   | 0.00                   | 0.00                  |
> | Chinese (zh)    | -                      | 0.00                   | 0.06                  |
> | Arabic (ar)     | 2.63                   | 1.49                   | 0.09                  |
> | Thai (th)       | 1.58                   | 0.00                   | 0.00                  |
> | Vietnamese (vi) | 1.13                   | 0.00                   | 0.00                  |
>
> **Llama-3.1-8B-Instruct:**
>
> | Source Language | CS to Chinese (zh) (%) | CS to Russian (ru) (%) | CS to Korean (ko) (%) |
> | --------------- | ---------------------- | ---------------------- | --------------------- |
> | English (en)    | 0.11                   | 0.00                   | 0.00                  |
> | Chinese (zh)    | -                      | 0.06                   | 0.28                  |
> | Arabic (ar)     | 0.87                   | 0.44                   | 0.09                  |
> | Thai (th)       | 1.14                   | 0.00                   | 0.00                  |
> | Vietnamese (vi) | 1.04                   | 0.00                   | 0.00                  |
>
> **Key Observations:**
>
> 1. **Arabic (ar)** is prone to code-switch to Chinese, Russian, and Korean across all models
> 2. **Thai (th)** and **Vietnamese (vi)** frequently code-switch to Chinese.
> 3. **Chinese (zh)** tends to code-switch to Korean.

---

> ### Author Response · Authors · 2025-11-20
> **Rebuttal (Part 4/4)**
>
> > Q5: Why compare with RL (GRPO), it is more likely to make alignment during the RL phase, not for multi-lingual extension.
>
> We compare with GRPO because it is the solution proposed by DeepSeek-AI in their technical report [1] for addressing code-switching in multilingual models. We follow their approach to provide a fair comparison against this established method.
>
> [1] DeepSeek-AI et al. DeepSeek-R1: Incentivizing Reasoning Capability in LLMs via Reinforcement Learning. arXiv:2501.12948
>
> > Q6: According to Table2, the proposed method does not show a significant improvement in effectiveness.
>
> The primary purpose of Table 2 is to demonstrate that **SASFT significantly reduces code-switching while maintaining overall capability across benchmarks**. Our goal is not to improve multilingual performance, but rather to mitigate code-switching without degrading existing model capabilities. As shown in Table 2, SASFT achieves substantial code-switching reduction, while keeping general performance stable across various tasks, which validates the effectiveness of our approach.
>
> > Q7: Figure 6, pls give detailed description for multi-layer, like the specific number of layers and which layer.
>
> In Figure 6, layers are indexed in reverse order where **0 represents the final (last) layer, 1 represents the second-to-last layer, and so on**.
>
> - **Single-layer** (solid lines): SASFT is applied to one individual layer at a time
> - **Multi-layer** (dashed lines): SASFT is applied to consecutive layers starting from the final layer up to the current layer. For example, at position 3, multi-layer means applying SASFT the last 4 layers.

---

> > ### Comment · Reviewer_aWvq · 2025-11-26
> >
> > Thank the authors for the clarifications provided. While these responses helped address several of my earlier concerns, certain aspects of the work still do not fully meet the threshold for acceptance in my opinion. Therefore, I decided to keep my original score, but I appreciate the authors’ efforts in improving the paper.

---

> > > ### Author Response · Authors · 2025-11-26
> > >
> > > Dear Reviewer,
> > >
> > > Thank you for taking the time to read our rebuttal and provide your response. We sincerely appreciate your acknowledgment of our efforts to improve the paper.
> > >
> > > We respect your decision to maintain the original score and understand that certain aspects may not yet fully meet your expectations. To help us better address your concerns and improve the paper's quality, would you be willing to briefly clarify which specific aspects remain as your main concerns?
> > >
> > > Any additional guidance you could provide would be extremely valuable for strengthening our work.
> > >
> > > Thank you again for your consideration.

---

### Official Review · Reviewer_iweR · 2025-10-31

**Soundness:** 3
**Presentation:** 3
**Contribution:** 3
**Rating:** 6
**Confidence:** 3

**Summary:**

This paper proposes a novel method, SASFT, to mitigate the issue of unexpected code-switching in large language models (LLMs). The method is grounded in Sparse Autoencoders (SAEs), which are used to identify language-specific feature. An auxiliary loss is introduced during supervised fine-tuning to suppress pre-activation values of irrelevant language features, thereby reducing unexpected code-switching. The method is evaluated on five multilingual LLMs across six benchmarks and three code-switching target languages, and shows strong reduction in code-switching rates, while mostly maintaining or improving performance on standard benchmarks.

**Strengths:**

1. The paper addresses an underexplored yet practically important problem, unexpected code-switching, which impacts user experience and model usability.

2. Demonstrates consistent reductions in code-switching across various models and languages, outperforming previous methods (e.g., GRPO) in most settings.

3. The paper offers detailed analysis on factors such as layer depth and feature selection.

4. The paper is clearly written and easy to follow.

**Weaknesses:**

1. The method for identifying language-specific features relies on rankings without justification. This introduces sensitivity to hyperparameter selection and limits the interpretability of the results, especially in multilingual settings where features may vary across tasks.

2. The paper reports a substantial +327% increase in Korean code-switching under the GRPO method, but does not provide sufficient explanation for this anomaly. A deeper analysis is needed to clarify the cause of such a drastic change.

3. It remains unclear why SASFT underperforms on certain benchmarks, such as MMLU and HellaSwag, for the Qwen3-8B model. This raises questions about robustness and generalizability across tasks.

**Questions:**

1. How does the proposed method perform on multilingual mathematical reasoning tasks, such as MGSM？

---

> ### Author Response · Authors · 2025-11-20
> **Rebuttal (Part 1/3)**
>
> We are grateful for your constructive feedback and the time you have dedicated to reviewing our work. We provide our detailed responses to each of your comments below. If you require any clarification, we are glad to provide additional information promptly.
>
> > The method for identifying language-specific features relies on rankings without justification. This introduces sensitivity to hyperparameter selection and limits the interpretability of the results, especially in multilingual settings where features may vary across tasks.
>
> Thank you for this important question about our feature identification methodology.
>
> Our method is inspired by Deng et al. [1], which provides an intuitive approach for discovering language-specific features in LLMs. To verify the robustness of this ranking-based approach, we applied the same identification method to SAE with different hyperparameters. Specifically, we computed the cosine similarity between rank #0 language feature vectors identified from Gemma Scope SAEs with varying sparsity (l0) and width. **The results below show that language features remain highly similar across different hyperparameters, with most similarities exceeding 0.85 and many above 0.90.** This demonstrates that our method reliably identifies consistent language-specific features regardless of SAE hyperparameters. More detailed results are provided in Appendix I.
>
> **Rank #0 Chinese Feature Similarity Across SAEs (Layer 24)**
>
> | Different SAEs       | l0_38 width_16k | l0_34 width_65k | l0_73 width_16k | l0_63 width_65k | l0_158 width_16k | l0_124 width_65k |
> | -------------------- | --------------- | --------------- | --------------- | --------------- | ---------------- | ---------------- |
> | **l0_38 width_16k**  | 1.0000          | 0.9084          | 0.9372          | 0.9253          | 0.8709           | 0.9043           |
> | **l0_34 width_65k**  | 0.9084          | 1.0000          | 0.8803          | 0.9467          | 0.8187           | 0.8984           |
> | **l0_73 width_16k**  | 0.9372          | 0.8803          | 1.0000          | 0.9236          | 0.9211           | 0.9281           |
> | **l0_63 width_65k**  | 0.9253          | 0.9467          | 0.9236          | 1.0000          | 0.8689           | 0.9518           |
> | **l0_158 width_16k** | 0.8709          | 0.8187          | 0.9211          | 0.8689          | 1.0000           | 0.9150           |
> | **l0_124 width_65k** | 0.9043          | 0.8984          | 0.9281          | 0.9518          | 0.9150           | 1.0000           |
>
> [1] Deng et al. Unveiling Language-Specific Features in Large Language Models via Sparse Autoencoders. ACL 2025

---

> ### Author Response · Authors · 2025-11-20
> **Rebuttal (Part 2/3)**
>
> > The paper reports a substantial +327% increase in Korean code-switching under the GRPO method, but does not provide sufficient explanation for this anomaly. A deeper analysis is needed to clarify the cause of such a drastic change.
>
> Thank you for this important observation. We believe this anomaly stems from inherent instabilities in the RL-based approach:
>
> **1. Length Sensitivity Issue:** Following DeepSeek, the reward is computed as the percentage of target language words in the output. This ratio-based metric is sensitive to response length - when models generate longer responses, the same amount of code-switching results in a higher reward. **The model may exploit this by generating longer text rather than genuinely reducing code-switching.**
>
> **2. Multi-language Code-Switching Confusion:** Within the same training batch, code-switching may occur to different languages. For example, Arabic prompts frequently trigger code-switching to Chinese, Korean, or Russian. When computing advantages across samples with code-switching to different languages, some "CS to language A" samples may receive positive advantages relative to "CS to language B" samples. **This causes the model to learn preferring one code-switching pattern over another, rather than avoiding code-switching entirely.**
>
> Our method does not suffer from these issues because it directly intervenes on the mechanistic level through SAE features, providing a more stable and interpretable solution without relying on reward hacking or relative comparisons.
>
> To better validate the effectiveness of our method, we propose **a more intuitive SFT baseline.** The main idea is to **add a penalty term to the SFT loss that minimizes the probabilities of tokens from a certain language.**
>
> Specifically, to reduce code-switch to a target language L (e.g., Chinese), we first identify all tokens belonging to language L from the tokenizer, denoted as $V_L$. During training on samples in languages other than L, in addition to the standard cross-entropy loss $L_{CE}$, we add a regularization term at each token position in the response to penalize the model's prediction probability for tokens in $V_L$. The training objective can be formulated as:
>
> $L_{TLP} = L_{CE} + \beta \cdot \frac{1}{T} \sum_{t=1}^{T} \sum_{v \in V_L} p_{\theta}(v|x,y_{ < t})$
>
> where:
>
> - $T$ is the response length
> - $p_{\theta}(v|x,y_{ < t})$ is the model's probability of predicting token $v$ at response position $t$
> - $\beta$ is the penalty coefficient
>
> We refer to this baseline as **SFT+Penalty**. We conducted hyperparameter search for the penalty coefficient $\beta$ and observed that SFT+Penalty often outperforms SFT+GRPO. However, **SASFT consistently outperforms both SFT+Penalty and SFT+GRPO across nearly all settings.** These results demonstrate that operating on feature-level representations provides more effective and consistent control over code-switching behavior than direct token-level penalization. The complete comparison has been added to Table 1 in the revised manuscript.
>
>
> | Model           | Method      | Data 210k  |            |            | Data 110k  |            |            |
> | --------------- | ----------- | ---------- | ---------- | ---------- | ---------- | ---------- | ---------- |
> |                 |             | CS: any→zh | CS: any→ru | CS: any→ko | CS: any→zh | CS: any→ru | CS: any→ko |
> | Gemma-2-2B      | SFT         | 0.82       | 0.35       | 3.78       | 0.55       | 0.58       | 1.26       |
> |                 | SFT+GRPO    | 0.70       | 0.49       | 3.35       | 0.58       | 0.35       | 1.16       |
> |                 | SFT+Penalty | 0.61       | 0.44       | 1.41       | 0.52       | 0.32       | 0.91       |
> |                 | SASFT       | **0.29**   | **0.09**   | **0.77**   | **0.32**   | **0.12**   | **0.35**   |
> | Llama-3.1-8B    | SFT         | 1.37       | 0.93       | 0.74       | 0.46       | 0.61       | 0.22       |
> |                 | SFT+GRPO    | 0.93       | 0.73       | 0.52       | 0.49       | 0.48       | 0.94       |
> |                 | SFT+Penalty | 0.49       | 0.67       | 0.49       | 0.38       | 0.41       | 0.37       |
> |                 | SASFT       | **0.26**   | **0.35**   | **0.37**   | **0.17**   | **0.26**   | **0.15**   |
> | Qwen3-1.7B-Base | SFT         | 0.46       | 0.15       | 0.22       | 0.55       | 0.15       | 0.22       |
> |                 | SFT+GRPO    | 0.73       | 0.12       | 0.27       | 0.47       | 0.15       | 0.12       |
> |                 | SFT+Penalty | 0.52       | 0.15       | 0.17       | 0.49       | 0.09       | 0.20       |
> |                 | SASFT       | **0.17**   | **0.06**   | **0.00**   | **0.18**   | **0.03**   | **0.02**   |

---

> ### Author Response · Authors · 2025-11-20
> **Rebuttal (Part 3/3)**
>
> > It remains unclear why SASFT underperforms on certain benchmarks, such as MMLU and HellaSwag, for the Qwen3-8B model. This raises questions about robustness and generalizability across tasks.
>
> **We believe these variations are within a reasonable range.** As shown in Appendix H, Qwen3-8B's performance on MMLU and HellaSwag varies across different experimental settings, **with both increases and decreases observed.** **We consider these to be reasonable fluctuations as long as the differences are not substantially large.** The key finding is that SASFT significantly reduces code-switching while maintaining overall capability across benchmarks.
>
> > How does the proposed method perform on multilingual mathematical reasoning tasks, such as MGSM？
>
> Thank you for this important question about multilingual mathematical reasoning capabilities.
>
> We add MGSM (Multilingual Grade School Math) evaluation to our experiments and include the results in the paper. Some results are shown in the table below:
>
> **MGSM Performance Comparison**
>
> | Model        | SFT   | SFT+GRPO      | SFT+Penalty   | SASFT         |
> | ------------ | ----- | ------------- | ------------- | ------------- |
> | Gemma-2-2B   | 12.05 | 10.99 (-1.06) | 11.97 (-0.08) | 12.24 (+0.19) |
> | Gemma-2-9B   | 44.61 | 45.84 (+1.23) | 46.35 (+1.74) | 44.96 (+0.35) |
> | Llama-3.1-8B | 19.92 | 22.83 (+2.91) | 15.81 (-4.11) | 18.35 (-1.57) |
> | Qwen3-1.7B   | 32.91 | 32.67 (-0.24) | 33.60 (+0.69) | 30.85 (-2.06) |
> | Qwen3-8B     | 58.03 | 55.28 (-2.75) | 56.29 (-1.74) | 58.45 (+0.42) |
>
> **SASFT maintains competitive performance on MGSM,** with improvements on Gemma-2-2B and Qwen3-8B. Minor variations on other models are within reasonable ranges, demonstrating that SASFT effectively reduces code-switching while preserving multilingual mathematical reasoning capabilities.

---

### Official Review · Reviewer_fhn1 · 2025-10-31

**Soundness:** 3
**Presentation:** 3
**Contribution:** 3
**Rating:** 6
**Confidence:** 5

**Summary:**

This paper addresses the problem of unexpected code-switching in multilingual large language models (LLMs). The authors first employ Sparse Autoencoders (SAEs) to analyze the internal representations of LLMs and discover that unexpected language switches are correlated with over-activation of target-language-specific features. Based on this finding, they propose a novel fine-tuning method, SASFT (Sparse Autoencoder-guided Supervised Finetuning), which introduces an auxiliary loss term during supervised fine-tuning to constrain the pre-activation values of specific language features below a threshold, thereby reducing activations of irrelevant languages. Experiments on five models and three target languages (Chinese, Russian, and Korean) show an average >50% reduction in unexpected switching, including four cases of complete elimination, while maintaining or improving performance on six multilingual benchmarks.

Main Contributions:
- Using sparse autoencoders, the paper reveals that when a model is about to unexpectedly switch to language L, the language-specific features of L show significantly elevated pre-activation values in the residual stream.
- Proposes SASFT, a training-stage approach that suppresses the activation of irrelevant language features without requiring inference-time intervention.
- Provides comprehensive experiments demonstrating that SASFT is effective, robust, and preserves multilingual capabilities.

**Strengths:**

1. The paper proposes a Sparse Autoencoder-guided Supervised Finetuning (SASFT) approach that combines sparse autoencoders with supervised fine-tuning.
2. The effectiveness of SASFT is validated through extensive experiments across multiple language pairs and model families. The results demonstrate consistent mitigation of code-switching phenomena in diverse multilingual settings.
3. Experimental evidence indicates that the proposed method effectively reduces unintended language switches, thereby improving the accuracy, consistency, and usability of multilingual model outputs.

**Weaknesses:**

1. Limited Baseline Comparison：The paper only compares SASFT with GRPO, which, although relevant, is insufficient to establish the method’s relative advantage.
2. Lack of Mechanistic or Causal Analysis：While the paper empirically observes that the pre-activation values of target-language features increase prior to code-switching and validates this via directional ablation, this evidence remains correlational. The work does not provide a mechanistic explanation of why such activation leads to language switching, nor tests the causal hypothesis.
3. Potential Model-Specific Bias. The reported improvements may partially stem from inherent differences in model multilingual balance, rather than from SASFT’s general efficacy. The paper does not control for or analyze how such model-specific language priors influence the observed reductions in code-switching.

**Questions:**

1. Could the authors provide a more comprehensive comparison to strengthen the empirical validity of SASFT?
2. The paper suggests that increased pre-activation of language-specific features precedes unexpected code-switching. Have the authors examined whether artificially increasing the activation of a non-target language feature can induce code-switching?

---

> ### Author Response · Authors · 2025-11-20
> **Rebuttal (Part 1/2)**
>
> Thank you for your thoughtful review. We sincerely appreciate your time and effort for the reviewing. We provide our point-by-point response to your concerns below. If anything is unclear, please let us know and we will reply promptly.
>
> > W1: Limited Baseline Comparison：The paper only compares SASFT with GRPO, which, although relevant, is insufficient to establish the method’s relative advantage.
> >
> > Q1: Could the authors provide a more comprehensive comparison to strengthen the empirical validity of SASFT?
>
> To better validate the effectiveness of our method, we propose **a more intuitive SFT baseline.** The main idea is to **add a penalty term to the SFT loss that minimizes the probabilities of tokens from a certain language.**
>
> Specifically, to reduce code-switch to a target language L (e.g., Chinese), we first identify all tokens belonging to language L from the tokenizer, denoted as $V_L$. During training on samples in languages other than L, in addition to the standard cross-entropy loss $L_{CE}$, we add a regularization term at each token position in the response to penalize the model's prediction probability for tokens in $V_L$. The training objective can be formulated as:
>
> $L_{TLP} = L_{CE} + \beta \cdot \frac{1}{T} \sum_{t=1}^{T} \sum_{v \in V_L} p_{\theta}(v|x,y_{ < t})$
>
> where:
>
> - $T$ is the response length
> - $p_{\theta}(v|x,y_{ < t})$ is the model's probability of predicting token $v$ at response position $t$
> - $\beta$ is the penalty coefficient
>
> We refer to this baseline as **SFT+Penalty**. We conducted hyperparameter search for the penalty coefficient $\beta$ and observed that SFT+Penalty often outperforms SFT+GRPO. However, **SASFT consistently outperforms both SFT+Penalty and SFT+GRPO across nearly all settings.** These results demonstrate that operating on feature-level representations provides more effective and consistent control over code-switching behavior than direct token-level penalization. The complete comparison has been added to Table 1 in the revised manuscript.
>
>
> | Model           | Method      | Data 210k  |            |            | Data 110k  |            |            |
> | --------------- | ----------- | ---------- | ---------- | ---------- | ---------- | ---------- | ---------- |
> |                 |             | CS: any→zh | CS: any→ru | CS: any→ko | CS: any→zh | CS: any→ru | CS: any→ko |
> | Gemma-2-2B      | SFT         | 0.82       | 0.35       | 3.78       | 0.55       | 0.58       | 1.26       |
> |                 | SFT+GRPO    | 0.70       | 0.49       | 3.35       | 0.58       | 0.35       | 1.16       |
> |                 | SFT+Penalty | 0.61       | 0.44       | 1.41       | 0.52       | 0.32       | 0.91       |
> |                 | SASFT       | **0.29**   | **0.09**   | **0.77**   | **0.32**   | **0.12**   | **0.35**   |
> | Llama-3.1-8B    | SFT         | 1.37       | 0.93       | 0.74       | 0.46       | 0.61       | 0.22       |
> |                 | SFT+GRPO    | 0.93       | 0.73       | 0.52       | 0.49       | 0.48       | 0.94       |
> |                 | SFT+Penalty | 0.49       | 0.67       | 0.49       | 0.38       | 0.41       | 0.37       |
> |                 | SASFT       | **0.26**   | **0.35**   | **0.37**   | **0.17**   | **0.26**   | **0.15**   |
> | Qwen3-1.7B-Base | SFT         | 0.46       | 0.15       | 0.22       | 0.55       | 0.15       | 0.22       |
> |                 | SFT+GRPO    | 0.73       | 0.12       | 0.27       | 0.47       | 0.15       | 0.12       |
> |                 | SFT+Penalty | 0.52       | 0.15       | 0.17       | 0.49       | 0.09       | 0.20       |
> |                 | SASFT       | **0.17**   | **0.06**   | **0.00**   | **0.18**   | **0.03**   | **0.02**   |
>
> > W3: Potential Model-Specific Bias. The reported improvements may partially stem from inherent differences in model multilingual balance, rather than from SASFT’s general efficacy. The paper does not control for or analyze how such model-specific language priors influence the observed reductions in code-switching.
>
> Thank you for raising this important concern about potential model-specific bias.
>
> Since existing open-source models do not disclose their pre-training language distributions, we cannot fully control for language priors. However, we use **five models from three different families** (Gemma, Llama, Qwen) and **two experimental settings (210k and 110k) with different language composition ratios** to demonstrate generalizability. The consistent effectiveness of SASFT across these diverse models and data settings provides evidence that our method's efficacy is not dependent on model-specific language priors.

---

> > ### Author Response · Authors · 2025-11-20
> > **Rebuttal (Part 2/2)**
> >
> > > W2: Lack of Mechanistic or Causal Analysis：While the paper empirically observes that the pre-activation values of target-language features increase prior to code-switching and validates this via directional ablation, this evidence remains correlational. The work does not provide a mechanistic explanation of why such activation leads to language switching, nor tests the causal hypothesis.
> > >
> > > Q2: The paper suggests that increased pre-activation of language-specific features precedes unexpected code-switching. Have the authors examined whether artificially increasing the activation of a non-target language feature can induce code-switching?
> >
> > To address this concern, **we validate that artificially increasing the activation of a target language feature can induce code-switching.** Specifically, we performed directional enhancement (the inverse of the ablation experiments in Section 3.3.2) by adding language features to the residual stream. We tested this on 100 samples that originally contained no code-switching to Chinese across five models. The results are shown in the table below.
> >
> > **Code-Switching Induction via Feature Enhancement** **(Metric: CS ratio %)**
> >
> > | Model      | Feature | λ=0  | λ=128 | λ=256 | λ=384 | λ=512 | λ=640 |
> > | ---------- | ------- | ---- | ----- | ----- | ----- | ----- | ----- |
> > | Gemma-2-2B | English | 0    | 1     | 0     | 0     | 0     | 0     |
> > | Gemma-2-2B | Chinese | 0    | 0     | 1     | 4     | 15    | 23    |
> >
> > | Model        | Feature | λ=0  | λ=16 | λ=32 | λ=64 | λ=80 | λ=96 | λ=112 |
> > | ------------ | ------- | ---- | ---- | ---- | ---- | ---- | ---- | ----- |
> > | Llama-3.1-8B | English | 0    | 0    | 0    | 0    | 0    | 0    | 0     |
> > | Llama-3.1-8B | Chinese | 0    | 0    | 2    | 25   | 47   | 68   | 86    |
> >
> > Enhancing Chinese features induces code-switching to Chinese, while enhancing English features produces negligible effect. While establishing rigorous causality is mathematically challenging in complex neural systems, these intervention-based results, combined with our ablation experiments, provide bidirectional evidence: both enhancing and reducing language-specific feature activations can induce and suppress code-switching respectively. **This strongly supports our mechanistic hypothesis that language-specific feature activation plays a determining role in language selection.** Complete experimental details and visualizations are provided in Appendix J.

---

### Official Review · Reviewer_dtNW · 2025-11-10

**Soundness:** 3
**Presentation:** 3
**Contribution:** 3
**Rating:** 6
**Confidence:** 3

**Summary:**

The paper studies unexpected code-switching in multilingual LLMs and links the phenomenon to unusually high pre-activation on language-specific SAE features. Building on this observation, it proposes SASFT: add an auxiliary loss during SFT that penalizes pre-activation of irrelevant language features across several layers.

**Strengths:**

1. Clear story from SAE analysis to a concrete training modification.
2.  The results are good to show the proposed method's effectiveness.
3. The paper is well-writen and easy to follow.

**Weaknesses:**

1. GRPO is run with only 10k samples (1k per language). That seems light for a control-behavior objective. Consider stronger RL baselines , or simple supervised baselines that directly penalize language-ID tokens . Without stronger baselines, it’s hard to attribute gains purely to SASFT. Does the author could ensure the GRPO have true convergence?
2. The study focuses on zh/ru/ko. It’s unclear if results hold for low-resource scripts (e.g., Amharic, Khmer), closely-related Latin languages where CS is subtler (es/pt/fr/it)?
3. SASFT relies on high-quality SAEs and on accurate language-feature identification; for Qwen the authors train their own SAEs, for others they reuse published ones. How sensitive are results to (1)SAE training corpora, dimensionality, sparsity target? (2)Which layer(s) the features are extracted from? (3) Feature drift after fine-tuning (do features stay monosemantic)?

**Questions:**

Please refer to the weakness.

---

> ### Author Response · Authors · 2025-11-20
> **Rebuttal  (Part 1/3)**
>
> We greatly appreciate your insightful feedback and the effort you have invested in evaluating our submission. Below is our point-by-point response to address your concerns. If there is any misunderstanding about your questions, please let us know, and we will respond ASAP.
>
> > GRPO is run with only 10k samples (1k per language). That seems light for a control-behavior objective. Consider stronger RL baselines , or simple supervised baselines that directly penalize language-ID tokens . Without stronger baselines, it’s hard to attribute gains purely to SASFT. Does the author could ensure the GRPO have true convergence?
>
> To better validate the effectiveness of our method, we propose **a more intuitive SFT baseline.** The main idea is to **add a penalty term to the SFT loss that minimizes the probabilities of tokens from a certain language.**
>
> Specifically, to reduce code-switch to a target language L (e.g., Chinese), we first identify all tokens belonging to language L from the tokenizer, denoted as $V_L$. During training on samples in languages other than L, in addition to the standard cross-entropy loss $L_{CE}$, we add a regularization term at each token position in the response to penalize the model's prediction probability for tokens in $V_L$. The training objective can be formulated as:
>
> $L_{TLP} = L_{CE} + \beta \cdot \frac{1}{T} \sum_{t=1}^{T} \sum_{v \in V_L} p_{\theta}(v|x,y_{ < t})$
>
> where:
>
> - $T$ is the response length
> - $p_{\theta}(v|x,y_{ < t})$ is the model's probability of predicting token $v$ at response position $t$
> - $\beta$ is the penalty coefficient
>
> We refer to this baseline as **SFT+Penalty**. We conducted hyperparameter search for the penalty coefficient $\beta$ and observed that SFT+Penalty often outperforms SFT+GRPO. However, **SASFT consistently outperforms both SFT+Penalty and SFT+GRPO across nearly all settings.** These results demonstrate that operating on feature-level representations provides more effective and consistent control over code-switching behavior than direct token-level penalization. The complete comparison has been added to Table 1 in the revised manuscript.
>
>
> | Model           | Method      | Data 210k  |            |            | Data 110k  |            |            |
> | --------------- | ----------- | ---------- | ---------- | ---------- | ---------- | ---------- | ---------- |
> |                 |             | CS: any→zh | CS: any→ru | CS: any→ko | CS: any→zh | CS: any→ru | CS: any→ko |
> | Gemma-2-2B      | SFT         | 0.82       | 0.35       | 3.78       | 0.55       | 0.58       | 1.26       |
> |                 | SFT+GRPO    | 0.70       | 0.49       | 3.35       | 0.58       | 0.35       | 1.16       |
> |                 | SFT+Penalty | 0.61       | 0.44       | 1.41       | 0.52       | 0.32       | 0.91       |
> |                 | SASFT       | **0.29**   | **0.09**   | **0.77**   | **0.32**   | **0.12**   | **0.35**   |
> | Llama-3.1-8B    | SFT         | 1.37       | 0.93       | 0.74       | 0.46       | 0.61       | 0.22       |
> |                 | SFT+GRPO    | 0.93       | 0.73       | 0.52       | 0.49       | 0.48       | 0.94       |
> |                 | SFT+Penalty | 0.49       | 0.67       | 0.49       | 0.38       | 0.41       | 0.37       |
> |                 | SASFT       | **0.26**   | **0.35**   | **0.37**   | **0.17**   | **0.26**   | **0.15**   |
> | Qwen3-1.7B-Base | SFT         | 0.46       | 0.15       | 0.22       | 0.55       | 0.15       | 0.22       |
> |                 | SFT+GRPO    | 0.73       | 0.12       | 0.27       | 0.47       | 0.15       | 0.12       |
> |                 | SFT+Penalty | 0.52       | 0.15       | 0.17       | 0.49       | 0.09       | 0.20       |
> |                 | SASFT       | **0.17**   | **0.06**   | **0.00**   | **0.18**   | **0.03**   | **0.02**   |

---

> ### Author Response · Authors · 2025-11-20
> **Rebuttal (Part 2/3)**
>
> > The study focuses on zh/ru/ko. It’s unclear if results hold for low-resource scripts (e.g., Amharic, Khmer), closely-related Latin languages where CS is subtler (es/pt/fr/it)?
>
> 1. **High-resource languages rarely code-switch to low-resource scripts in practice.** As shown in the table below, we tested five open-source models on approximately 10,000 samples across ten high-resource languages (English, Chinese, Arabic, Spanish, French, Japanese, Korean, Portuguese, Thai, and Vietnamese), measuring code-switching rates to three low-resource scripts: Khmer (km), Lao (lo), and Burmese (my). All models exhibited zero code-switching to these low-resource languages across all source languages. This is likely because models have weaker proficiency in low-resource languages and tend to rely on better-learned high-resource languages for expression instead, while the reverse does not happen.
>
> | Model         | CS ratio to km | CS ratio to lo | CS ratio to my |
> | ------------- | -------------- | -------------- | -------------- |
> | Gemma-2-2b-it | 0%             | 0%             | 0%             |
> | Gemma-2-9b-it | 0%             | 0%             | 0%             |
> | Llama-3.1-8B  | 0%             | 0%             | 0%             |
> | Qwen3-1.7B    | 0%             | 0%             | 0%             |
> | Qwen3-8B      | 0%             | 0%             | 0%             |
>
> 2. **Same-script code-switching lacks reliable evaluation tools.** We did not investigate code-switching among same-script languages (e.g., Spanish/Portuguese/English) due to the lack of reliable evaluation methods. Existing tools cannot detect fine-grained code-switching, such as single characters in another language [1]. We therefore employ a writing-system-based detection approach using Unicode character ranges, which cannot differentiate between languages sharing the same script (e.g., Latin). We believe developing robust evaluation methods for same-script code-switching is an important direction that we plan to explore in future work.
>
> [1] Burchell et al. Code-switched language identification is harder than you think. EACL'2024.

---

> ### Author Response · Authors · 2025-11-20
> **Rebuttal (Part 3/3)**
>
> > SASFT relies on high-quality SAEs and on accurate language-feature identification; for Qwen the authors train their own SAEs, for others they reuse published ones. How sensitive are results to (1)SAE training corpora, dimensionality, sparsity target? (2)Which layer(s) the features are extracted from? (3) Feature drift after fine-tuning (do features stay monosemantic)?
>
> 1. **Language-specific features are highly robust across different SAE hyperparameters.** We computed the cosine similarity between rank #0 language feature vectors identified from Gemma Scope SAEs with varying sparsity (l0) and width. The tables below show that language features remain highly similar across different hyperparameters, with most similarities exceeding 0.85 and many above 0.90. This demonstrates that our method reliably identifies consistent language-specific features regardless of SAE hyperparameters. More detailed results across layers 19-25 are provided in Appendix I.
>
> **Rank #0 Chinese Feature Similarity Across SAEs (Layer 24)**
>
> | Different SAEs       | l0_38 width_16k | l0_34 width_65k | l0_73 width_16k | l0_63 width_65k | l0_158 width_16k | l0_124 width_65k |
> | -------------------- | --------------- | --------------- | --------------- | --------------- | ---------------- | ---------------- |
> | **l0_38 width_16k**  | 1.0000          | 0.9084          | 0.9372          | 0.9253          | 0.8709           | 0.9043           |
> | **l0_34 width_65k**  | 0.9084          | 1.0000          | 0.8803          | 0.9467          | 0.8187           | 0.8984           |
> | **l0_73 width_16k**  | 0.9372          | 0.8803          | 1.0000          | 0.9236          | 0.9211           | 0.9281           |
> | **l0_63 width_65k**  | 0.9253          | 0.9467          | 0.9236          | 1.0000          | 0.8689           | 0.9518           |
> | **l0_158 width_16k** | 0.8709          | 0.8187          | 0.9211          | 0.8689          | 1.0000           | 0.9150           |
> | **l0_124 width_65k** | 0.9043          | 0.8984          | 0.9281          | 0.9518          | 0.9150           | 1.0000           |
>
> 2. In our main results, we apply SASFT using language features from **the last two layers.** We thoroughly investigate the impact of layer selection in Section 5.3. We extracted language features from the final layer up to the 8th layer from the end, comparing single-layer versus multi-layer approaches. As shown in Figure 6, multi-layer SASFT consistently outperforms single-layer approaches across all models and demonstrates more stable performance.
>
> 3. **Feature drift after fine-tuning is not significant:** Recent research has shown that SAE features generally transfer well between base and chat models [1]. To verify this for language-specific features, we applied the same SAE and feature identification method to both Gemma-2-2B base and instruction-tuned models across 10 languages. The table below shows the overlap rate of identified language features:
>
>    | Layer | Rank #0 Feature Overlap (%) | Rank #1 Feature Overlap (%) |
>    | ----- | --------------------------- | --------------------------- |
>    | 18    | 100                         | 95                          |
>    | 19    | 100                         | 95                          |
>    | 20    | 100                         | 95                          |
>    | 21    | 100                         | 80                          |
>    | 22    | 100                         | 95                          |
>    | 23    | 100                         | 80                          |
>    | 24    | 100                         | 90                          |
>    | 25    | 100                         | 85                          |
>
>    The results show 100% overlap for rank #0 features and high overlap (80-95%) for rank #1 features, indicating that language-specific features remain stable during fine-tuning and supporting the reliability of our approach.
>
>    [1] https://www.alignmentforum.org/posts/fmwk6qxrpW8d4jvbd/saes-usually-transfer-between-base-and-chat-models

---

### Meta-Review · Area_Chair_xt1D · 2026-01-08

**Summary:**

The authors introduced Sparse Autoencoder-guided Supervised Finetuning (SASFT), a new method to teach LLMs to maintain pre-activation values accordingly to language features. The authors conducted experiments with SASFT and found that it can reduce unexpected code-switching significantly compared to standard SFT.

**Reviewer Concerns:**

- The baseline of using GRPO is not sufficient because of limited training data and weaker training signals.
    - The authors conducted additional experiments with stronger GRPO and also with SFT + penalty loss for code switching. The results showed that SASFT still outperforms these baselines.
- The paper explored popular languages (zh, ru, ko) and did not demonstrate on low-resource languages or closely-related Latin languages (es/pt/etc.)
    - The authors addressed this concern by arguing that high-resource languages rarely code-switch to low-resource scripts with reported statistics
- Concerns on the causality between code switching and pre-activation values. The paper did not provide a mechanistic explanation to explain this causality. Because of this, model-specific bias (e.g. language priors) might also impact the results and findings.

**Reviewer Scores:**

- Reviewer dtNW would keep the same positive score of 6
- Reviewer fhn1 would keep the same positive score of 6
- Reviewer iweR might increase the score from 6 to 7 given their concerns are addressed
- Reviewer aWvq might keep the same score of 4. However, some of the authors’ concerns are not well supported due to potential misunderstanding of the paper.

---

### Decision · Program_Chairs · 2026-01-26

Accept (Poster)